# Dynamical Field Inference and Supersymmetry

**DOI:** 10.3390/e23121652

**Published:** 2021-12-08

**Authors:** Margret Westerkamp, Igor Ovchinnikov, Philipp Frank, Torsten Enßlin

**Affiliations:** 1Max Planck Institute for Astrophysics, Karl-Schwarzschildstraße 1, 85748 Garching, Germany; 2Physics Department, Ludwig-Maximilians-Universität, Geschwister-Scholl Platz 1, 80539 Munich, Germany; 3Excellence Cluster Universe, Technische Universität München, Boltzmannstr. 2, 85748 Garching, Germany

**Keywords:** information field theory, field inference, supersymmetric theory of stochastics, stochastic differential equations, chaos theory

## Abstract

Knowledge on evolving physical fields is of paramount importance in science, technology, and economics. Dynamical field inference (DFI) addresses the problem of reconstructing a stochastically-driven, dynamically-evolving field from finite data. It relies on information field theory (IFT), the information theory for fields. Here, the relations of DFI, IFT, and the recently developed supersymmetric theory of stochastics (STS) are established in a pedagogical discussion. In IFT, field expectation values can be calculated from the partition function of the full space-time inference problem. The partition function of the inference problem invokes a functional Dirac function to guarantee the dynamics, as well as a field-dependent functional determinant, to establish proper normalization, both impeding the necessary evaluation of the path integral over all field configurations. STS replaces these problematic expressions via the introduction of fermionic ghost and bosonic Lagrange fields, respectively. The action of these fields has a supersymmetry, which means there exists an exchange operation between bosons and fermions that leaves the system invariant. In contrast to this, measurements of the dynamical fields do not adhere to this supersymmetry. The supersymmetry can also be broken spontaneously, in which case the system evolves chaotically. This affects the predictability of the system and thereby makes DFI more challenging. We investigate the interplay of measurement constraints with the non-linear chaotic dynamics of a simplified, illustrative system with the help of Feynman diagrams and show that the Fermionic corrections are essential to obtain the correct posterior statistics over system trajectories.

## 1. Introduction

Stochastic differential equations (SDEs) appear in many disciplines like astrophysics [1], biology [2], chemistry [3], and economics [4,5]. In contrast to traditional differential equations the dynamics of the system, which follows the SDE, are influenced by initial and boundary conditions but not entirely determined by them. The uncertainty in the dynamics can be an intrinsic stochastic behavior [6] or simply due to imperfections in the model [7], which describes the dynamical system (DS).

In addition to the uncertainty introduced by the stochastic process driving the evolution of the system, any observation of it is noise afflicted and incomplete. This complicates the inference of the system’s state further. In previous studies, linear SDEs [8], especially the Langevin SDE [9], were already investigated extensively. Besides this, many numerical methods to solve partial differential equations were interpreted and the propagation of the uncertainty for these problems has been studied [10,11]. Here, we consider arbitrary SDEs and introduce dynamical field inference (DFI) as a Bayesian framework to estimate the state and evolution of a field following a SDE from finite, incomplete, and noise-afflicted data. DFI rests on information field theory (IFT), which is information theory for fields. IFT [12,13] was developed in order to be able to reconstruct an infinite dimensional signal from some finite dimensional data, as the signal from physical reality is usually not limited to the discrete space. Rather a physical signal is described by a continuous signal field. In contrast, the data taken from a measurement can never be continuous. IFT can then be applied for signal inference in all areas, where limitations on the exactness of the measurement are given. DFI [14,15,16] utilizes methods from IFT for the inference of signals in a DS. The reconstruction of the signal is advanced by the knowledge on the signal properties, which are specified by the prior covariance of the signal. Non-linearities in the SDE result in a complicated and signal-dependent structure of the covariance. The central mathematical object of our investigation will be the partition function of the inference problem, from which any relevant quantity of interest can be obtained. The importance of the partition function for the calculation of dynamical critical properties was also outlined in [17]. This partition function is represented by a path integral involving a functional delta function, to enforce the system dynamics, and a functional determinant, to ensure proper normalization of the involved probability densities. To handle the delta function and determinant, bosonic Lagrange and fermionic ghost fields are respectively introduced.

The approach of supersymmetric theory of stochastics (STS) focuses on the theoretical analysis of the DS as a supersymmetric system [18,19,20,21]. One of the central messages of stochastics (STS) is the correspondence between the spontaneous breakdown of this supersymmetry (SUSY) and the emergence of chaotic dynamics. Here, we argue that the emergence of chaos impacts the ability to infere dynamical fields. The dynamical growth rates of the fermionic ghost fields, which are the Lyapuov coefficients measuring the strength of chaos, impact the uncertainty of any field inference. Thereby, we illuminate the relevance of central elements of STS for DFI.

The paper tries to give a pedagogical introduction into IFT and STS by presenting the elementary calculation steps in all derivations. The paper is structured as follows: In Section 2, a brief introduction to IFT is given, from which, the perspective DFI is developed in Section 3. Bosonic Lagrange and fermionic ghost fields are introduced in Section 4. These permit for a reformulation of the partition function such that a symmetry between all bosonic and fermionic degrees of freedom becomes apparent. Section 5.1 investigates the relation between SUSY and DFI by showing that system measurements have no SUSY and how spontaneously broken SUSY, which was already investigated in [22], aka chaos impacts field reconstructions from measurement data. In Section 5.2 and Section 5.3, we analyze the impact of the chaos on the predictability for linear and non-linear dynamic. With having connected the DFI and STS formalisms, and shown their mutual relevances, we conclude in Section 6 and give an outlook on future research directions.

## 2. Information Field Theory

In many areas of science, technology, and economics, the difficult task of interpreting incomplete and noisy data sets and computing the uncertainty of the results arises [23,24]. If the quantity of interest is a field, for example, a spatially extended component of our Galaxy [25,26], or of the atmosphere [27,28], which are mostly continous functions over a physical space, the problem becomes virtually infinte dimensional, as any point in space-time carries one or several degrees of freedom. For such problems, which are called field inference problems, IFT was developed. IFT can be considered as a combination of information theory for distributed quantities and statistical field theory.

### 2.1. Notation

Usually, only certain aspects describing our system ψ are relevant. These aspects are called the signal, φ. Physical degrees of freedom, which are contained in ψ and not in φ, but which still influence the data, are called noise *n*. If φ is a physical field φ:Ω→R, it is a function that assigns a value to each point in time and *u*-dimensional position space. Let us denote a space-time location by $x=\left(\vec{x},t\right)\in \Omega =R^{u}\times R_{0}^{+}$, u∈N, where space and time will be handled in the same manner initially as in [29,30]. We let the time axis start at $t_{0}=0$ for definiteness.

The field φ=φ(x) has an infinite number of degrees of freedom and integrations over the phase space of the field are represented by path integrals over the integration measure $D\varphi =\prod_{x\in \Omega }d\varphi_{x}$ [31], with $\varphi_{x}=\varphi \left(x\right)$ being a more compact notation. In the following, these space-time coordinate dependent fields are denoted as abstract vectors in Hilbert space. The scalar product between two fields φ(x) and γ(x) can be written in short notation as:(1)$$\begin{array}{c} \gamma^{†}\varphi :=\int dx\;\gamma^{*}\left(x\right)\varphi \left(x\right), \end{array}$$
where $\gamma^{*}$ is the complex conjugate of γ, which here will play no role, as we deal only with real valued fields.

### 2.2. Bayesian Updating

In order to get to know a field φ, one has to measure it. Bayes theorem states how to update any existing knowledge given a finite number of constraints by measurements that resulted in the data vector *d*. Apparently, it is not possible to reconstruct the infinite dimensional field configuration of φ perfectly from a finite number of measurements. This is where the probabilistic description used in IFT comes into play. In probabilistic logic, knowledge states are described by probability distributions.

After the measurement of data *d*, the knowledge according to Bayes theorem [13] is given by the posterior probability distribution:(2)$$\begin{array}{c} P\left(\varphi |d\right)=\frac{P(d|\varphi )P(\varphi )}{P(d)}. \end{array}$$

This posterior is proportional to the likelihood P(d|φ) of the measured data given the signal field multiplied by the prior probability distribution P(φ). The normalization of the posterior is given by the so-called evidence:(3)$$\begin{array}{c} P(d)=\int D\varphi \;P(d|\varphi )\;P(\varphi ). \end{array}$$

Bayes theorem describes the update of knowledge states. The prior P(φ) turns into the posterior P(φ|d) given some data *d*. To construct the posterior, we need to have the prior and the likelihood. The evidence and posterior incorporate those.

### 2.3. Prior Knowledge

The prior probability of φ, P(φ), specifies the knowledge on the signal before any measurement was performed. Formally, the prior on φ can be written in terms of the system prior [12]:(4)$$\begin{array}{c} P(\varphi )=\int D\psi \;\delta (\varphi -\varphi (\psi ))\;P(\psi ), \end{array}$$
where φ(ψ) is the function that specifies the field φ given the system state ψ. Due to the integration over ψ, the underlying system becomes partly invisible in the probability densities and only the field of interest, the signal field φ, remains. Nevertheless, the properties of the original systems will still be present in the field prior P(φ). For example, let us consider a situation close to what will be relevant later on. We consider a system comprised of two interacting fields constituting the system ψ=(φ,η), which are related via the invertible functional G[φ]=η. This implies the conditional probability P(η|φ)=δ(η−G[φ]), which can be considered as a first-class constraint in Dirac’s sense [32]. Then we have, assuming that there exists a unique solution φ to the equation G[φ]=η,  
(5)$$\begin{array}{ccc} P(\eta |\varphi ) & = & \delta (\eta -G[\varphi ])\;\mathrm{and}\\ P(\varphi |\eta ) & = & \delta (\varphi -G^{-1}\left[\eta \right])\\ & = & \frac{\delta (\eta -G[\varphi ])}{||\delta G^{-1}\left[\eta \right]/\delta \eta \left|\right|} \end{array}$$
(6)$$\begin{array}{ccc} & = & ∥\frac{\delta G[\varphi ]}{\delta \varphi }∥\;\delta \left(\eta -G\left[\varphi \right]\right). \end{array}$$

We casted P(φ|η) into a form that only requires to have access to *G*, but not to $G^{-1}$. As *G* is one to one, $P\left(\varphi |\eta \right)=\delta (\varphi -G^{-1}\left(\eta \right))$ would be our preferred quantity to work with. However, in DFI of non-linear systems, we rarely have $G^{-1}$ available as an explicit expression and therefore have to restore to Equation (Equation 6). Now, we assume that we know the prior statistics of P(η) and find the following implications on P(φ),
(7)$$\begin{array}{cc} P(\varphi ) & =\int D\eta \;P(\varphi |\eta )P(\eta )\\ \; & =\int D\eta \;∥\frac{\delta G[\varphi ]}{\delta \varphi }∥\;\delta \left(\eta -G\left[\varphi \right]\right)\;P\left(\eta \right)\\ \; & =∥\frac{\delta G[\varphi ]}{\delta \varphi }∥\;P\left(\eta =G\left[\varphi \right]\right). \end{array}$$

This shows that the field of interest φ inherits the statistics of the related field η, however, with a modification by the functional determinant ||∂G/∂φ|| that is sensitive to non-linearities in the field relation. Here, the probability P(φ|η) contains already the two elements that will lead to SUSY in DFI, the delta function, which will be represented with bosonic Lagrange fields and the functional determinant, for which fermionic fields are introduced. Since both terms contain the functional *G*, it is plausible that bosons and fermions might be connected via a symmetry.

### 2.4. Likelihood

Let us now turn to the measurement and its likelihood. The measurement process of the data can always be written as:(8)$$\begin{array}{c} d=R[\varphi ]+n, \end{array}$$
if we define the signal response to be $R\left[\varphi \right]={\langle d\rangle }_{\left(d|\varphi \right)}:=\int Dd\;P\left(d|\varphi \right)\;d$ and the noise as n:=d−R[φ]. In measurement practice, the response converts a continuous signal into a discrete data set. The linear noise of the measurement is given by the residual vector in data space between data and signal response, n=d−R[φ]. The statistics of the noise, which can be signal dependent, then determines the likelihood,
(9)$$\begin{array}{ccc} P(d|\varphi ) & = & \int Dn\;P(d,n|\varphi )\;\\ & = & \int Dn\;P(d|n,\varphi )\;P(n|\varphi )\;\\ & = & \int Dn\;\delta (d-R[\varphi ]-n)\;P(n|\varphi )\\ & = & P(n=d-R[\varphi ]|\varphi ). \end{array}$$

Note, however, that we might want to specify initial conditions of a dynamical field via data as well. Let $\varphi_{0}=\varphi \left(\cdot ,t_{0}\right)$ be the initial field configuration at initial time $t_{0}$. Then, we specify the initial data to be exactly this initial field configuration, $d_{0}=\varphi_{0}$, the corresponding response as $R_{0}\left[\varphi \right]=\varphi \left(\cdot ,t_{0}\right),$ and the noise to vanish, P(n)=δ(n). Now, the initial condition is represented via the likelihood $P\left(d_{0}|\varphi \right):=P\left(d|\varphi ,\;d_{0}\;\;=\;\;\varphi \left(\cdot ,t_{0}\right)\right)=\delta \left(\varphi \left(\cdot ,t_{0}\right)-\varphi_{0}\right)$. This initial data likelihood can be combined with any other data on the later evolution, $d_{l}$, via $P\left(d|\varphi \right)=P\left(d_{0}|\varphi \right)\;P\left(d_{l}|\varphi \right)$, where $d=(d_{0},d_{l})$ is the combined data vector.

### 2.5. Information

Bayes theorem Equation (Equation 2) can be rewritten in terms of statistical mechanics by defining an information Hamiltonian, or short the information, which contains all the information needed for inference, and the partition function, which serves as a normalization factor,
(10)$$\begin{array}{cc} P(\varphi |d) & =\frac{e^{-H(d,\varphi )}}{Z_{d}}, \end{array}$$
(11)$$\begin{array}{cc} H(d,\varphi ) & :=-\ln(P(d,\varphi )), \end{array}$$
(12)$$\begin{array}{cc} Z_{d} & :=\int D\varphi \;e^{-H(d,\varphi )}. \end{array}$$

Note, these formal definitions of information Hamiltonian and partition function hold in the absence of a thermodynamic equilibrium. This formulation of field inference in terms of a statistical field theory permits the usage of the well-developed apparatus of field theory, as we briefly show in the following.

### 2.6. Partition Function

There is an infinite number of possible signal field realizations that meet the constraints given by a finite number of measurements as encoded in the field posterior P(φ|d). For practical purposes, for example to have a figure in a publication showing what is known about a field, one has to extract lower dimensional views of this very high dimensional posterior function. These can be obtained by calculating posterior expectation values of the signal field, like its posterior mean $m={\langle \varphi \rangle }_{\left(\varphi |d\right)}=\int D\varphi \;P\left(\varphi |d\right)\;\varphi$ or its uncertainty dispersion $D={\langle \left(\varphi -m\right)\;{\left(\varphi -m\right)}^{†}\rangle }_{\left(\varphi |d\right)}$. Thus, we want to be able to calculate posterior field moments.

Given some data on a signal field φ, the posterior *n*-point function is:(13)$$\begin{array}{c} {\langle \varphi \left(x_{1}\right)...\varphi \left(x_{n}\right)\rangle }_{\left(\varphi |d\right)}=\int D\varphi \;\varphi \left(x_{1}\right)\;...\;\varphi \left(x_{n}\right)\;P\left(\varphi |d\right). \end{array}$$

The involved integral can be calculated exactly in case the posterior P(φ|d) is a Gaussian. Otherwise, the posterior may be expanded around a Gaussian.

With the help of the moment generating function:(14)$$\begin{array}{c} Z_{d}\left[J\right]=\int D\varphi \;e^{-H\left(d,\varphi \right)+J^{†}\varphi }, \end{array}$$
which incorporates a moment generating source term $J^{†}\varphi =\int dxJ^{*}\left(x\right)\varphi \left(x\right)$, the moments can be calculated via derivation with respect to *J* as:(15)$$\begin{array}{c} {\langle \varphi \left(x_{1}\right)..\varphi \left(x_{n}\right)\rangle }_{\left(\varphi |d\right)}:=\frac{1}{Z_{d}\left[J\right]}\frac{\delta^{n}Z_{d}\left[J\right]}{\delta J^{*}\left(x_{1}\right)...\delta J^{*}\left(x_{n}\right)}{|}_{J=0}. \end{array}$$

Likewise, the connected correlation functions, also called cumulants, are defined as:(16)$$\begin{array}{c} {\langle \varphi \left(x_{1}\right)..\varphi \left(x_{n}\right)\rangle }_{\left(\varphi |d\right)}^{c}:=\frac{\delta^{n}\log Z_{d}\left[J\right]}{\delta J^{*}\left(x_{1}\right)...\delta J^{*}\left(x_{n}\right)}{|}_{J=0}. \end{array}$$

Particularly, the cumulants of the first and second order are of importance as they describe the posterior mean and uncertainty dispersion, $m={\langle \varphi \rangle }_{\left(\varphi |d\right)}^{c}={\langle \varphi \rangle }_{\left(\varphi |d\right)}^{}$ and $D={\langle \varphi \;\varphi {}^{†}\rangle }_{\left(\varphi |d\right)}^{c}={\langle \left(\varphi -m\right)\;{\left(\varphi -m\right)}^{†}\rangle }_{\left(\varphi |d\right)}$, respectively. Thus, the ultimate goal of any field inference is to obtain the moment generating partition function $Z_{d}\left[J\right]$ as any desired *n*-point correlation function can be calculated from it. For this reason, this partition function will be the focus of our investigations.

### 2.7. Free Theory

An illustrative example for the signal reconstruction and the simplest scenario in IFT is given by the free theory. The underlying initial assumptions of the free theory lead to a theory without non-linear field interactions. In other words, the information H(d,φ) includes no terms of an order higher than quadratic in the signal field φ.

The free theory emerges in practice under the following conditions:(i)A Gaussian zero-centered prior, P(φ)=G(φ,Φ), with known covariance $\Phi ={\langle \varphi \varphi^{†}\rangle }_{\left(\varphi \right)}$;(ii)A linear measurement, d=Rφ+n, with known linear response *R* and additive noise;(iii)A signal-independent Gaussian noise, P(n|φ)=G(n,N), with known covariance $N={\langle nn^{†}\rangle }_{\left(n\right)}$.

The information H(d,φ) is then calculated via the data likelihood and the signal prior,
(17)$$\begin{array}{c} H(d,\varphi )=-\log(P(d|\varphi ))-\log(P(\varphi )). \end{array}$$

With the assumptions of the free theory and Equation (Equation 9) the likelihood is:(18)$$\begin{array}{cc} P(d|\varphi ) & =G(R\;\varphi -d,N). \end{array}$$

Thus, the information for the free theory is given by:(19)$$\begin{array}{cc} H(d,\varphi ) & =-\log(G(R\varphi -d,N)\;G(\varphi ,\Phi ))\\ \; & =\frac{1}{2}\varphi^{†}\left(R^{†}N^{-1}R+\Phi^{-1}\right)\varphi -d^{†}N^{-1}R\varphi \end{array}$$(20)$$\begin{array}{cc} \; & \;\;\;\;+\frac{1}{2}\ln\left(|2\pi N|\right)+\frac{1}{2}\ln\left(|2\pi \Phi |\right)+\frac{1}{2}d^{†}N^{-1}d \end{array}$$(21)$$\begin{array}{cc} \; & =\frac{1}{2}\varphi^{†}D^{-1}\varphi -j^{†}\varphi +H_{0}. \end{array}$$

Here, the so-called information source *j*, the information propagator *D*, and $H_{0}$ were introduced. The latter contains all the terms of the information that are constant in φ. The others are,
(22)$$\begin{array}{cc} D & ={\left(\Phi^{-1}+R^{†}N^{-1}R\right)}^{-1}, \end{array}$$
(23)$$\begin{array}{cc} \; & =\Phi -\Phi \;R^{†}{\left(R\;\Phi \;R^{†}+N\right)}^{-1}R\;\Phi \end{array}$$
(24)$$\begin{array}{cc} j & =R^{†}N^{-1}d. \end{array}$$

The second form of the information propagator *D* can be verified via explicit calculation,  
(25)$$\begin{array}{ccc} & & D\;D^{-1}\\ & = & \left[\Phi -\Phi \;R^{†}{\left(R\;\Phi \;R^{†}+N\right)}^{-1}R\;\Phi \right]\;\left[\Phi^{-1}+R^{†}N^{-1}R\right]\\ & = & \left[𝟙-\Phi \;R^{†}{\left(R\;\Phi \;R^{†}+N\right)}^{-1}R\right]\;\left[𝟙+\Phi \;R^{†}N^{-1}R\right]\\ & = & 𝟙+\Phi \;R^{†}N^{-1}R-\Phi \;R^{†}{\left(R\;\Phi \;R^{†}+N\right)}^{-1}R\\ & & -\Phi \;R^{†}{\left(R\;\Phi \;R^{†}+N\right)}^{-1}R\;\Phi \;R^{†}N^{-1}R\\ & = & 𝟙+\Phi \;R^{†}N^{-1}R-\Phi \;R^{†}{\left(R\;\Phi \;R^{†}+N\right)}^{-1}R\\ & & -\Phi \;R^{†}{\left(R\;\Phi \;R^{†}+N\right)}^{-1}\left(R\;\Phi \;R^{†}+N-N\right)N^{-1}R\\ & = & 𝟙+\Phi \;R^{†}N^{-1}R-\Phi \;R^{†}{\left(R\;\Phi \;R^{†}+N\right)}^{-1}R\\ & & -\Phi \;R^{†}N^{-1}R+\Phi \;R^{†}{\left(R\;\Phi \;R^{†}+N\right)}^{-1}R\\ & = & 𝟙 \end{array}$$
and also holds in the limit N→0 of a noise-less measurement.

The information can be expressed in terms of the field:(26)m=Dj
by completing the square in Equation (Equation 21), which is also known as the generalized Wiener filter solution [33]. This can also be written in a form that permits a noiseless measurement limit,
(27)$$\begin{array}{ccc} m & = & {\left(\Phi^{-1}+R^{†}N^{-1}R\right)}^{-1}R^{†}N^{-1}d\\ & = & R^{†}\Phi \;\left(R\;\Phi \;R^{†}+N\right)d, \end{array}$$
which can be verified with a very analogous calculation.

Only terms, which depend on the signal field φ need to be considered and therefore the symbol “$\hat{=}$” is introduced, to mark the equality up to an additive constant. We therefore have:(28)$$\begin{array}{c} H\left(d,\varphi \right)\;\hat{=}\;\frac{1}{2}{\left(\varphi -m\right)}^{†}D^{-1}\left(\varphi -m\right). \end{array}$$

Knowing the information, the moment generating function of the free theory, $Z_{G}\left[J\right]$, is constructed in the next step on the way of calculating the best fit reconstruction of the signal by means of expectation values.
(29)$$\begin{array}{cc} Z_{G}\left[J\right] & =\int D\varphi \;e^{-H\left(d,\varphi \right)+J^{†}\varphi } \end{array}$$
(30)$$\begin{array}{cc} \; & =\sqrt{\left|2\pi D\right|}e^{\frac{1}{2}{\left(j+J\right)}^{†}D\left(j+J\right)-H_{0}}. \end{array}$$

All higher order (n>2) cumulants vanish and the non-vanishing cumulants are,
(31)$$\begin{array}{cc} m(x) & ={\langle \varphi \left(x\right)\rangle }_{\left(\varphi |d\right)}^{c}=\frac{\delta \log Z_{G}\left[J\right]}{\delta J^{*}\left(x\right)}{|}_{J=0}, \end{array}$$
(32)$$\begin{array}{cc} D(x,y) & ={\langle \varphi \left(x\right)\varphi^{*}\left(y\right)\rangle }_{\left(\varphi |d\right)}^{c}=\frac{\delta^{2}\log Z_{G}\left[J\right]}{\delta J^{*}\left(x\right)\delta J\left(y\right)}{|}_{J=0}. \end{array}$$

As higher-order cumulants vanish, the posterior distribution can be written as a Gaussian with mean *m* and uncertainty covariance *D*,
(33)$$\begin{array}{c} P(\varphi |d)=G(\varphi -m,D). \end{array}$$

Hence, computations in free theory are simple, as the Gaussian posterior can be treated analytically. The usage of the same symbol *D* for the information propagator, the inverse of the kernel of the quadratic term in the information, and the posterior uncertainty dispersion is justified, as they coincide in the free theory, but only there.

In other cases, when the signal or noise are non-Gaussian, the response non-linear or the noise is signal dependent, the theory becomes interacting in the sense that H(d,φ) contains terms that are of higher than quadratic order. Thus, the information of this non-free, interacting theory incorporates not only the propagator and source terms of the free theory but also interaction terms between more than two signal field values. We will encounter such situations for a field with non-linear dynamics.

## 3. Dynamical Field Inference

### 3.1. Field Prior

In the previous section, we saw how to infer a signal field from measurement data *d* with some measurement noise *n* particularly in the case of a free theory. Now, we consider a DS, for which the time evolution of the signal field is described by an SDE:(34)$$\begin{array}{c} \partial_{t}\varphi \left(x\right)=F\left[\varphi \right]\left(x\right)+\xi \left(x\right). \end{array}$$

We want to see how this knowledge can be incorporated into a prior for the field for DFI. The first part of the SDE in Equation (Equation 34), $\partial_{t}\varphi \left(x\right)=F\left[\varphi \right]\left(x\right)$, describes the deterministic dynamics of the field. The excitation field ξ turns the deterministic evolution into an SDE and mirrors the influence of external factors on the dynamics. DFI aims to infer a signal in such a DS using the tools from IFT. Thus, in DFI next to the observational *n*, which results from the measurement contaminated by nuisance influences, the excitation field ξ of the SDE has to be considered during inference.

Care has to be taken as the domains of the fields φ and ξ differ. While φ(x) is defined far all $x\in \Omega =R^{u}\times R_{0}^{+}$, the fields $\partial_{t}\varphi$ and ξ live only over $\Omega^{\prime }=R^{u}\times R^{+}$, from which the intial time slice at $t_{0}=0$ is removed. Equation (Equation 34) therefore makes only statements about fields on $\Omega^{\prime }$, although it also depends on the intial conditions $\varphi_{0}=\varphi \left(\cdot ,t_{0}\right).$ As such requires specification, an initial condition prior $P(\varphi_{0})$ is required. We further introduce the notation $\varphi^{\prime }=\varphi \left(\cdot ,t\neq t_{0}\right)$ for all field degrees of freedom except the ones fixed by the initial condition, $\varphi_{0}$, so that we have $\varphi =(\varphi_{0},\varphi^{\prime })$.

The SDE in Equation (Equation 34) can be condensed and generalized by a differential operator G[φ], $G:C^{n,1}\left(\Omega \right)\to C\left(\Omega^{\prime }\right)$, which contains all the time and space derivatives of the SDE up to order *n* in space. In other words, the operator *G* acts on the space $C^{n,1}$ which is the class of all functions that have continuous first derivatives in time and continuous n-th derivatives in space.
(35)$$\begin{array}{cc} G[\varphi ](x) & =\xi (x)\;\mathrm{with} \end{array}$$
(36)$$\begin{array}{cc} G[\varphi ](x) & :=\partial_{t}\varphi \left(x\right)-F\left[\varphi \left(\cdot ,t\right)\right]\left(x\right). \end{array}$$

Within the framework of this study, we will assume that the excitation of the SDE has a prior Gaussian statistics,
(37)$$\begin{array}{cc} P(\xi ) & =G(\xi ,\Xi ), \end{array}$$
with known covariance Ξ. For a general *G*, ξ in its present form does not fully specify φ, for this additional initial conditions $\varphi_{0}$ at time $t_{0}$ have to be specified. We fix this by augmenting ξ with: $\varphi_{0}=\varphi \left(\cdot ,t_{0}\right)$ by setting $\eta ={\left(\varphi_{0},\xi \right)}^{†}$ with
(38)$$P\left(\eta \right)=P\left(\varphi_{0}\right)\;G\left(\xi ,\Xi \right),$$
and by extending *G* to:(39)$$\begin{array}{ccc} G^{\prime }\left[\varphi \right] & = & \left(\varphi_{0},G\left[\varphi \right]\right) \end{array}$$
with $G^{\prime }:C^{n,1}\left(\Omega \right)\to C\left(\Omega \right)$ such that $G^{\prime }\left[\varphi \right]=\eta$ and $G^{\prime -1}\left[\eta \right]=\varphi$ hold and are both uniquely defined.

Then, the prior probability for the signal field is according to Equation (Equation 6),
(40)$$\begin{array}{cc} P(\varphi ) & =P\left(\eta =G^{\prime }\left[\varphi \right]\right)\;∥\frac{\delta G^{\prime }\left[\varphi \right]}{\delta \varphi }∥\\ \; & =G\left(G\left[\varphi \right],\Xi \right)\;P\left(\varphi_{0}\right)\;∥\frac{\delta G^{\prime }\left[\varphi \right]}{\delta \varphi }∥, \end{array}$$
and the functional determinant becomes:(41)$$\begin{array}{ccc} ∥\frac{\delta G^{\prime }\left[\varphi \right]}{\delta \varphi }∥ & = & ∥\left(\begin{array}{cc} \frac{\delta \varphi_{0}}{\delta \varphi_{0}} & \frac{\delta G[\varphi ]}{\delta \varphi_{0}}\\ \frac{\delta \varphi_{0}}{\delta \varphi^{\prime }} & \frac{\delta G[\varphi ]}{\delta \varphi^{\prime }} \end{array}\right)∥=∥\left(\begin{array}{cc} 𝟙 & \frac{\delta G[\varphi ]}{\delta \varphi_{0}}\\ 0 & \frac{\delta G[\varphi ]}{\delta \varphi^{\prime }} \end{array}\right)∥\\ & = & ∥\frac{\delta G[\varphi ]}{\delta \varphi^{\prime }}∥, \end{array}$$
where we note that $\delta G/\delta \varphi^{\prime }:C^{n,1}\left(\Omega \right)\times C\left(\Omega^{\prime }\right)\to C\left(\Omega^{\prime }\right)$ and therefore, after evaluation of this for a specific field configuration φ, $\delta G\left[\varphi \right]/\delta \varphi^{\prime }:C\left(\Omega^{\prime }\right)\to C\left(\Omega^{\prime }\right)$ is a linear operator, which actually is an isomorphism. Thus, we get finally:(42)$$\begin{array}{cc} P(\varphi ) & =G\left(G\left[\varphi \right],\Xi \right)\;P\left(\varphi_{0}\right)\;∥\frac{\delta G[\varphi ]}{\delta \varphi^{\prime }}∥. \end{array}$$

If we want to have the initial conditions unconstrained, we could set $P(\varphi_{0})=\mathrm{const}$. This is possible, as we could specify initial or later time conditions via additional data on the field, as explained before.

### 3.2. Partition Function

DFI builds on P(d,φ)=P(d|φ)P(φ), the joint probability of data and field, to obtain field expectation values by investigating the moment generating partition function:(43)$$\begin{array}{ccc} Z_{d}\left[J\right] & = & \int D\varphi \;P\left(d,\varphi \right)\;e^{J^{†}\varphi }\\ & = & \int D\varphi \;P\left(d|\varphi \right)\;P\left(\varphi \right)\;e^{J^{†}\varphi }\\ & = & \int D\varphi \;\frac{e^{-\frac{1}{2}{\left(d-R\varphi \right)}^{†}N^{-1}\left(d-R\varphi \right)+J^{†}\varphi }}{\sqrt{\left|2\pi N\right|}}\;P\left(\varphi \right)\\ & = & \int D\varphi \;\frac{e^{-\frac{1}{2}\varphi^{†}R^{†}N^{-1}R\varphi +{\left(J+j\right)}^{†}\varphi -\frac{1}{2}d^{†}N^{-1}d}}{\sqrt{\left|2\pi N\right|}}\;P\left(\varphi \right)\\ \mathrm{with}\;j & = & R^{†}N^{-1}d. \end{array}$$

Here, we used that the measurement noise exhibits Gaussian statistics with known covariance *N*. We observe that the generating function *J* is not needed, as we could equally well take derivatives with respect to *j* in order to generate moments.

Central to this partition function is the field prior:(44)$$\begin{array}{ccc} P(\varphi ) & = & P\left(\xi =G\left[\varphi \right]\right)\;∥\frac{\delta G[\varphi ]}{\delta \varphi^{\prime }}∥P\left(\varphi_{0}\right) \end{array}$$(45)$$\begin{array}{ccc} & = & \underset{=:B(\varphi )}{\underbrace{\frac{1}{\sqrt{\left|2\pi \Xi \right|}}e^{-\frac{1}{2}G{\left[\varphi \right]}^{†}\Xi^{-1}G\left[\varphi \right]}}}\underset{=:J(\varphi )}{\underbrace{∥\frac{\delta G[\varphi ]}{\delta \varphi^{\prime }}∥}}P\left(\varphi_{0}\right). \end{array}$$

This contains a signal-dependent term B(φ) from the excitation statistics as well as another one, J(φ), from the functional determinant. In particular, the calculation of this determinant remains a computational problem. The aim of the next section is to represent the Jacobian determinant J by a path-integral over fermionic fields for the data-free partition function:(46)$$\begin{array}{cc} Z & =\int D\varphi \;P\left(\varphi \right)=\int D\varphi \;e^{-H(\varphi )}\\ \; & =\int D\varphi \;B\left(\varphi \right)\;J\left(\varphi \right)\;P\left(\varphi_{0}\right). \end{array}$$

## 4. Dynamical Field Inference with Ghost Fields

### 4.1. Grassmann Fields

Grassmann numbers $\left\{\chi_{1},{\bar{\chi }}_{1},\ldots \chi_{N},{\bar{\chi }}_{N}\right\}$ are independent elements, which anticommute among each other [34,35,36] and thus follow the Pauli principle, $\chi_{i}^{2}={\bar{\chi }}_{i}^{2}=0$ for i∈{1,…N}. Consequently, a corresponding function depending on the Grassmann numbers χ and $\bar{\chi }$ can be Taylor expanded to:(47)$$\begin{array}{c} f\left(\chi ,\bar{\chi }\right)=a+b_{1}\chi +b_{2}\bar{\chi }+c_{12}\chi \bar{\chi }+c_{21}\bar{\chi }\chi . \end{array}$$

A special feature of Grassmann numbers is that the integration and differentiation to them are the same. As a consequence, one can write down the following Grassmann integrals: (48)$$\begin{array}{cc} \int d\chi \;d\bar{\chi } & =0 \end{array}$$(49)$$\begin{array}{cc} \int d\chi \;d\bar{\chi }\;\bar{\chi }\chi & =1 \end{array}$$

In order to represent the Jacobian with infinite dimensions by a path integral, we need to transform the Grassmann variables to Grassmann fields with infinite dimensions. This leads us to path integrals over Grassmann fields,
(50)$$\begin{array}{c} \int d\chi_{1}d{\bar{\chi }}_{1}\;...\;d\chi_{N}d{\bar{\chi }}_{N}\overset{N\to \infty }{\overset{\;}{\to }}\int D\chi D\bar{\chi }, \end{array}$$
with the following integration rules,
(51)$$\begin{array}{cc} \; & \int D\chi \;D\bar{\chi }=0 \end{array}$$
(52)$$\begin{array}{cc} \; & \int D\chi \;D\bar{\chi }\;{\bar{\chi }}^{†}\chi =\int D\chi \;D\bar{\chi }\;\left(\int_{\Omega^{\prime }}dx\bar{\chi }\left(x\right)\chi \left(x\right)\right)=𝟙, \end{array}$$
where the ${\bar{\chi }}^{†}$ is the adjoint of the anti-commuting field $\bar{\chi }$. The scalar product:(53)$${\bar{\chi }}^{†}\chi =\int_{\Omega^{\prime }}\;\;dx\;\bar{\chi }\left(x\right)\chi \left(x\right)$$
will here be taken only over the domain $\Omega^{\prime }$ without the inital time slice, as the Grassmann fields are introduced to represent a determinant of the functional J(φ), which is also defined only over this domain. In the following, we abbreviate the notation by writing ∫dx for $\int_{\Omega^{\prime }}dx$.

### 4.2. Path Integral Representation of Determinants and δ-Functions

By means of the Grassmann fields, we derive the path integral representation for J, the absolute value of the determinant of the Jacobian $\frac{\delta G[\varphi ]}{\delta \varphi^{\prime }}$ [37]. For this purpose, we take two unitary transformations *U* and *V* with the property that $M=V\frac{\delta G[\varphi ]}{\delta \varphi^{\prime }}U$ becomes diagonal with positive and real entries. These are then used to transform the Grassmann fields:(54)$$\chi =U\chi^{\prime },\;\;{\bar{\chi }}^{†}={\bar{\chi }}^{\prime †}V.$$

This leads to a weighting of the path integral differentials by the determinants of *U* and *V*:(55)$$D\chi D\bar{\chi }={\left|U\right|}^{-1}{\left|V\right|}^{-1}D\chi^{\prime }D{\bar{\chi }}^{\prime }.$$

Here we used the identity of integration and differentiation for Grassmann variables $\int d\chi =\frac{\partial }{\partial \chi }=\frac{\partial \chi^{\prime }}{\partial \chi }\frac{\partial }{\partial \chi^{\prime }}={\left|U\right|}^{-1}\int d\chi^{\prime }$ to transform their differentials. The determinant of the operator *M* is given by the product of the operators, from which we can infer the Jacobian determinant:(56)$$\begin{array}{cc} \left|M\right| & =|V|\;\left|\frac{\delta G[\varphi ]}{\delta \varphi^{\prime }}\right|\;\left|U\right| \end{array}$$(57)$$\begin{array}{cc} \Rightarrow \left|\frac{\delta G[\varphi ]}{\delta \varphi^{\prime }}\right| & ={\left|M|\;|U\right|}^{-1}\;{\left|V\right|}^{-1}. \end{array}$$

As the operator *M* is diagonal with eigenvalues $\left\{m_{i}\right\}$ on the diagonal, we can write its determinant as a product of *N* eigenvalues in the limit of infinite dimensions *N* by means of Equations (Equation 47)–(Equation 49).
(58)$$\begin{array}{cc} \left|M\right| & =\lim_{N\to \infty }\prod_{i=1}^{N}m_{i}\\ \; & =\lim_{N\to \infty }\prod_{i=1}^{N}\left[\underset{=0}{\underbrace{\int d\chi_{i}^{\prime }\;d{\bar{\chi }}_{i}^{\prime }}}+m_{i}\underset{=1}{\underbrace{\int d\chi_{i}^{\prime }\;d{\bar{\chi }}_{i}^{\prime }\;{\bar{\chi }}_{i}^{\prime }\chi_{i}^{\prime }}}\right]\\ \; & =\lim_{N\to \infty }\prod_{i=1}^{N}\int d\chi_{i}^{\prime }\;d{\bar{\chi }}_{i}^{\prime }\;\left(1+m_{i}{\bar{\chi }}_{i}^{\prime }\chi_{i}^{\prime }\right)\\ \; & =\lim_{N\to \infty }\prod_{i=1}^{N}\int d\chi_{i}^{\prime }\;d{\bar{\chi }}_{i}^{\prime }\;\left(1+m_{i}{\bar{\chi }}_{i}^{\prime }\chi_{i}^{\prime }+\underset{=0}{\underbrace{\frac{1}{2!}m_{i}^{2}{\left({\bar{\chi }}_{i}^{\prime }\chi_{i}^{\prime }\right)}^{2}}}\right)\\ \; & =\lim_{N\to \infty }\prod_{i=1}^{N}\int d\chi_{i}^{\prime }\;d{\bar{\chi }}_{i}^{\prime }\;e^{m_{i}{\bar{\chi }}_{i}^{\prime }\chi_{i}^{\prime }}. \end{array}$$

The insertion of the result for the determinant of the diagonal matrix *M* in the definition of the Jacobian in Equation (Equation 57) using Equation (Equation 55) yields:  
(59)$$\begin{array}{cc} \left|\frac{\delta G[\varphi ]}{\delta \varphi^{\prime }}\right| & ={\left|U\right|}^{-1}\;{\left|V\right|}^{-1}\lim_{N\to \infty }\prod_{i=1}^{N}\int d\chi_{i}^{\prime }\;d{\bar{\chi }}_{i}^{\prime }\;e^{m_{i}{\bar{\chi }}_{i}^{\prime }\chi_{i}^{\prime }}\\ \; & =\int D\chi^{\prime }\;D{\bar{\chi }}^{\prime }{\;|U|}^{-1}\;{\left|V\right|}^{-1}e^{{\bar{\chi }}^{\prime †}M\chi^{\prime }}\\ \; & =\int D\chi^{\prime }\;D{\bar{\chi }}^{\prime }{\;|U|}^{-1}\;{\left|V\right|}^{-1}e^{{\bar{\chi }}^{†}V^{-1}MU^{-1}\chi }\\ \; & =\int D\chi D\bar{\chi }e^{{\bar{\chi }}^{†}\frac{\delta G[\varphi ]}{\delta \varphi^{\prime }}\chi }. \end{array}$$

Finally, we find the representation of the Jacobian in terms of an integral over independent Grassmann fields,
(60)$$\begin{array}{c} J=∥\frac{\delta G[\varphi ]}{\delta \varphi^{\prime }}∥=\left|\int D\chi D\bar{\chi }\;e^{{\bar{\chi }}^{†}\frac{\delta G[\varphi ]}{\delta \varphi^{\prime }}\chi }\right|. \end{array}$$

We note that an equivalent expression is:(61)$$\begin{array}{c} J=∥-i\frac{\delta G[\varphi ]}{\delta \varphi^{\prime }}∥=\left|\int D\chi \;D\bar{\chi }\;e^{-i{\bar{\chi }}^{†}\frac{\delta G[\varphi ]}{\delta \varphi^{\prime }}\chi }\right|, \end{array}$$
as the factor −i cancels out in taking the absolute value. In the following, we will not track such multiplicative factors of unity absolute value for probabilities, as these can be fixed at the end of the calculation.

The other term in $P\left(\varphi \right)=B\left(\varphi \right)\;J\left(\varphi \right)\;P\left(\varphi_{0}\right)$ as expressed by Equation (Equation 44), B(φ)=G(G[φ],Ξ), is highly non-Gaussian for a non-linear dynamics G[φ]. Here, it is useful to step back to the initial form including the excitation field:(62)$$B\left(\varphi \right)=\int D\xi \;\delta \left(\xi -G\left[\varphi \right]\right)\;e^{-H(\xi )},$$
with $H\left(\xi \right)=-\ln G\left(\xi ,\Xi \right)=\frac{1}{2}\xi^{†}\Xi^{-1}\xi +\frac{1}{2}\ln\left|2\pi \Xi \right|$, and to replace the δ-function by means of a path integral. In order to do so the representation of the δ-function as an integral over Fourier modes is recalled:(63)$$\begin{array}{c} \delta \left(x\right)=\frac{1}{2\pi }\int dk\;e^{-ikx}. \end{array}$$

The migration of this to path-integral representation is achieved by the introduction of a Lagrange multiplier field β(x),
(64)$$\begin{array}{cc} \delta (\xi ) & =\frac{1}{\left|2\pi 𝟙\right|}\int D\beta \;e^{-i\beta^{†}\xi }. \end{array}$$

With this, the field prior reads:(65)$$\begin{array}{ccc} P(\varphi ) & \propto & \int \frac{D\xi D\beta D\chi D\bar{\chi }}{\sqrt{\left|2\pi \Xi \right|}\left|2\pi 𝟙\right|}\;e^{-\frac{1}{2}\xi^{†}\Xi^{-1}\xi -H\left(\varphi_{0}\right)}\;\times \\ & & e^{-i\left({\bar{\chi }}^{†}\frac{\delta G[\varphi ]}{\delta \varphi^{\prime }}\chi -\beta^{†}\left(G\left[\varphi \right]-\xi \right)\right)} \end{array}$$
with $H\left(\varphi_{0}\right)=-\ln P\left(\varphi_{0}\right)$ the information on the initial conditions.

### 4.3. Ghost Field Path Integrals in DFI

With the introduction of the fields β, χ, and $\bar{\chi }$, the DFI partition function is now given by path integrals over the excitations and additional two fermionic and two bosonic degrees of freedom, which are summarized to a tuple of fields $\psi =(\varphi ,\;\beta ,\;\chi ,\;\bar{\chi })$, (note, the here defined ψ differs from the initially introduced system state, also denoted by ψ. As the latter will not be used any more in this work, the reuse of the symbol is hopefully acceptable).
(66)$$\begin{array}{c} Z\propto \int D\xi \;D\psi \;e^{-H\left(\xi \right)-H\left(\varphi_{0}\right)+i\beta^{†}\left(G\left[\varphi \right]-\xi \right)-i{\bar{\chi }}^{†}\frac{\delta G[\varphi ]}{\delta \varphi^{\prime }}\chi }. \end{array}$$

Let us introduce the functional {Q[ψ],·}={Q[χ,β],·}, which depends on the fermionic ghost field χ and the bosonic Lagrange multiplier β:(67)$$\begin{array}{cc} \left\{Q,X\}[\psi \right] & =\int dx\;\left(\beta \left(x\right)\frac{\delta }{\delta \bar{\chi }\left(x\right)}+\chi \left(x\right)\frac{\delta }{\delta \varphi^{\prime }\left(x\right)}\right)X\left[\psi \right]\\ \; & ={\left(\beta \frac{\delta }{\delta \bar{\chi }}+\chi \frac{\delta }{\delta \varphi^{\prime }}\right)}^{T}X\left[\psi \right]. \end{array}$$

Next, the exponent of the partition function in Equation (Equation 66) is reshaped in order to be *Q*-exact. This means that the exponent shall only depend on the introduced functional {Q,·} for a suitable *X*. For this we investigate the two ghost and Lagrange field dependent terms in Equation (Equation 66) separately.

The fermionic ghost field dependent exponent is:(68)$$\begin{array}{cc} E_{\mathrm{fg}} & =-i{\bar{\chi }}^{†}\frac{\delta G[\varphi ]}{\delta \varphi^{\prime }}\chi \\ \; & =-i\int dx^{\prime }dx\;\bar{\chi }\left(x\right)\frac{\delta G[\varphi ](x)}{\delta \varphi^{\prime }\left(x^{\prime }\right)}\chi \left(x^{\prime }\right)\\ \; & =i\int dx^{\prime }dx\;\chi \left(x^{\prime }\right)\frac{\delta }{\delta \varphi^{\prime }\left(x^{\prime }\right)}G\left[\varphi \right]\left(x\right)\bar{\chi }\left(x\right)\\ \; & =i\left(\chi^{†}\frac{\delta }{\delta \varphi^{\prime }}\right)\left({\bar{\chi }}^{†}\;G\left[\varphi \right]\right)\\ \; & =i\left(\chi^{†}\frac{\delta }{\delta \varphi^{\prime }}\right)\left({\bar{\chi }}^{†}\;\left(G\left[\varphi \right]-\xi \right)\right) \end{array}$$
and the bosonic Lagrange field dependent exponent is:(69)$$\begin{array}{cc} E_{\mathrm{bg}} & =i\beta^{†}\left(G\left[\varphi \right]-\xi \right)\\ \; & =i\int dx\;\beta (x)(G[\varphi ](x)-\xi (x))\\ \; & =i\int dx\;dx^{\prime }\;\beta \left(x^{\prime }\right)\frac{\delta \bar{\chi }\left(x\right)}{\delta \bar{\chi }\left(x^{\prime }\right)}\left(G\left[\varphi \right]\left(x\right)-\xi \left(x\right)\right)\\ \; & =i\left(\beta^{T}\frac{\delta }{\delta \bar{\chi }}\right)\left({\bar{\chi }}^{†}\;\left(G\left[\varphi \right]-\xi \right)\right). \end{array}$$

Thus the whole ghost and Lagrange field dependent exponent can be written as a *Q*-exact expression using Equations (Equation 68) and (Equation 69):         
(70)$$\begin{array}{ccc} E_{\mathrm{fg}}+E_{\mathrm{bg}} & = & i\beta^{†}\left(G\left[\varphi \right]-\xi \right)-i{\bar{\chi }}^{†}\frac{\delta G[\varphi ]}{\delta \varphi^{\prime }}\chi \\ & = & i\left(\chi^{†}\;\frac{\delta }{\delta \varphi^{\prime }}+\beta^{†}\;\frac{\delta }{\delta \bar{\chi }}\right)\left({\bar{\chi }}^{†}\;\left(G\left[\varphi \right]-\xi \right)\right)\\ & = & i\{Q,{\bar{\chi }}^{†}\left(G\left[\varphi \right]-\xi \right)\}. \end{array}$$

According to these auxiliary calculations, the partition function in Equation (Equation 66) takes the form,
(71)$$\begin{array}{cc} Z & \propto \int D\xi \;D\psi \;e^{-H\left(\xi \right)-H\left(\varphi_{0}\right)+i\left\{Q,{\bar{\chi }}^{†}\left(G\left[\varphi \right]-\xi \right)\right\}}. \end{array}$$

The integration over the excitation fields creates a partition function that only contains the fields of the set $\psi =(\varphi ,\;\beta ,\;\chi ,\;\bar{\chi })$. With the aid of the following relation for a bosonic field y(x) that is independent of φ:(72)$$\begin{array}{cc} \left\{Q,{\bar{\chi }}^{†}y\right\} & =\left(\beta^{†}\frac{\delta }{\delta \bar{\chi }}\right){\bar{\chi }}^{†}y\\ \; & =\int dx^{\prime }\;\beta \left(x^{\prime }\right)\frac{\delta }{\delta \bar{\chi }\left(x^{\prime }\right)}\int dx\;\bar{\chi }\left(x\right)y\left(x\right)\\ \; & =\int dx\;\beta (x)y(x)\\ \; & =\beta^{†}y \end{array}$$
the integration over the excitation field can be performed for a Gaussian excitation field ($H\left(\xi \right)\;\hat{=}\;\frac{1}{2}\xi^{†}\xi$) by means of Equation (Equation 72):(73)$$\begin{array}{cc} Z & \;\propto \int D\psi \;D\xi \;e^{i\left\{Q,{\bar{\chi }}^{†}G\left[\varphi \right]\right\}-i\left\{Q,{\bar{\chi }}^{†}\xi \right\}-H\left(\xi \right)-H\left(\varphi_{0}\right)}\\ \; & \overset{\left(72\right)}{=}\int D\psi \;D\xi \;e^{i\left\{Q,{\bar{\chi }}^{†}G\left[\varphi \right]\right\}-i\beta^{†}\xi -H\left(\xi \right)-H\left(\varphi_{0}\right)}\\ \; & \;=\int D\psi \;D\xi \;e^{i\left\{Q,{\bar{\chi }}^{†}G\left[\varphi \right]\right\}-i\beta^{†}\xi -\frac{1}{2}\xi^{†}\Xi^{-1}\xi -H\left(\varphi_{0}\right)}\\ \; & \;=\int D\psi \;e^{i\left\{Q,{\bar{\chi }}^{†}G\left[\varphi \right]\right\}-\frac{1}{2}\beta^{†}\Xi \beta -H\left(\varphi_{0}\right)}\\ \; & \;=\int D\psi \;e^{i\left\{Q,{\bar{\chi }}^{†}G\left[\varphi \right]\right\}-\frac{1}{2}\left\{Q,{\bar{\chi }}^{†}\Xi \beta \right\}-H\left(\varphi_{0}\right)}\\ \; & \;=\int D\psi \;e^{\left\{Q,i{\bar{\chi }}^{†}G\left[\varphi \right]-\frac{1}{2}{\bar{\chi }}^{†}\Xi \beta \right\}-H\left(\varphi_{0}\right)}. \end{array}$$

Now, we define the odd function:(74)$$\begin{array}{cc} \theta (\psi ) & ={\bar{\chi }}^{†}\left(-iG\left[\varphi \right]+\frac{1}{2}\Xi \beta \right) \end{array}$$
for reasons of clarity. Besides we revive the statistical mechanics formalism for the definition of the partition function from Equation (Equation 12) as well as the corresponding ghost and Lagrange field dependent information H(ψ): (75)$$\begin{array}{cc} Z & =\int D\psi \;e^{-H\left(\varphi_{0}\right)-H\left(\psi |\varphi_{0}\right)} \end{array}$$(76)$$\begin{array}{cc} \; & \propto \int D\psi \;e^{-H\left(\varphi_{0}\right)-\left\{Q,\theta \left(\psi \right)\right\}} \end{array}$$(77)$$\begin{array}{cc} H(\psi |\varphi_{0}) & \;\hat{=}\;\left\{Q,\theta \left(\psi \right)\right\}. \end{array}$$

Here, $\hat{=}$ indicates equality up to a constant term due to the not tracked absolute phase of our expressions. By comparison, we find the following relation between the prior information Hamiltonian of the signal field H(φ) from Equation (Equation 12) and the just derived information Hamiltonian of the ghost and Lagrange fields from Equation (Equation 75).
(78)$$\begin{array}{cc} Z & \propto \int D\varphi \;e^{-H\left(\varphi |\varphi_{0}\right)-H\left(\varphi_{0}\right)} \end{array}$$
(79)$$\begin{array}{cc} \; & =\int D\psi \;e^{-H\left(\psi |\varphi_{0}\right)-H\left(\varphi_{0}\right)} \end{array}$$
(80)$$\begin{array}{cc} \Rightarrow & H\left(\varphi |\varphi_{0}\right)=-\ln\left(\int D\chi \;D\bar{\chi }\;D\beta \;e^{-H(\psi |\varphi_{0})}\right). \end{array}$$

Let us now emphasize the first time derivative in the SDE by taking the definition of the SDE from Equation (Equation 34), $F\left[\varphi^{\prime }\right]\left(x\right)+\xi \left(x\right)=\partial_{t}\varphi \left(x\right)$, so that the θ-functional becomes:(81)$$\begin{array}{cc} \theta (\psi ) & =+i{\bar{\chi }}^{†}F\left[\varphi^{\prime }\right]-i{\bar{\chi }}^{†}\partial_{t}\varphi +\frac{1}{2}{\bar{\chi }}^{†}\Xi \beta \\ \; & =-i{\bar{\chi }}^{†}\partial_{t}\varphi +i\bar{Q}\left(\psi \right). \end{array}$$

Here we introduced the functional on the set of fields ψ:(82)$$\begin{array}{c} \bar{Q}\left(\psi \right)={\bar{\chi }}^{†}F\left[\varphi^{\prime }\right]-i\frac{1}{2}{\bar{\chi }}^{†}\Xi \beta . \end{array}$$

Evaluating the information for this θ-functional using Equation (Equation 77) one gets:(83)$$\begin{array}{cc} H(\psi |\varphi_{0}) & \hat{=}-\left\{Q,i{\bar{\chi }}^{†}\partial_{t}\varphi \right\}+i\left\{Q,\bar{Q}\right\}. \end{array}$$

The Fermionic field χ was only defined over $\Omega^{\prime }$ the field domain without the initial time slice in order to represent the determinant of the Jacobian of G(φ) with respect to $\varphi^{\prime }.$ One can extend the support of χ to Ω, including the initial time slice by introducing a split notation for this extended $\chi ={\left(\chi_{0},\chi^{\prime }\right)}^{†}$, with $\chi^{\prime }$ denoting the original Fermionic field over $\Omega^{\prime }$. We then find that the ghost field has to vanish at the initial time step $t_{0}$, i.e., $\chi ={\left(0,\chi^{\prime }\right)}^{†}$ in order to assure that the following expression does not diverge. Here, we abbreviate   $\varphi_{t}=\varphi \left(x\right)=\varphi \left(\vec{x},t\right)\;\mathrm{and}\;\varphi_{t+\Delta }=\varphi \left(\vec{x},t+\Delta t\right)$: (84)$$\begin{array}{cc} \; & \left\{Q,i{\bar{\chi }}^{†}\partial_{t}\varphi \right\}\\ = & \left(\chi^{†}\frac{\delta }{\delta \varphi^{\prime }}+\beta^{†}\frac{\delta }{\delta \bar{\chi }}\right)\;i{\bar{\chi }}^{†}\partial_{t}\varphi \\ = & \underset{=A}{\underbrace{i\chi^{†}\frac{\delta }{\delta \varphi^{\prime }}{\bar{\chi }}^{†}\partial_{t}\varphi }}+\underset{=B}{\underbrace{i\beta^{†}\frac{\delta }{\delta \bar{\chi }}{\bar{\chi }}^{†}\partial_{t}\varphi }} \end{array}$$(85)$$\begin{array}{cc} A= & i\int dx^{\prime }\;dx\;\chi \left(x^{\prime }\right)\;\chi \left(x\right)\frac{\delta }{\delta \varphi^{\prime }\left(x^{\prime }\right)}\partial_{t}\varphi \left(x\right)\\ = & -i\int dx\;dx^{\prime }\bar{\chi }\left(x\right)\;\chi \left(x^{\prime }\right)\frac{\delta }{\delta \varphi^{\prime }\left(x^{\prime }\right)}\lim_{\Delta t\to 0}\left(\begin{array}{c} \frac{\varphi_{0+\Delta }-\varphi_{0}}{\Delta t}\\ \frac{\varphi_{t+\Delta }-\varphi_{t}}{\Delta t} \end{array}\right)\\ = & -i\int dx\;dx^{\prime }\bar{\chi }\left(x\right)\;\chi \left(x^{\prime }\right)\;\lim_{\Delta t\to 0}\left(\begin{array}{c} \frac{\delta_{\vec{x},{\vec{x}}^{\prime }}\delta_{0+\Delta ,t^{\prime }}}{\Delta t}\\ \frac{\delta_{\vec{x},{\vec{x}}^{\prime }}\left(\delta_{t+\Delta ,t^{\prime }}-\delta_{t,t^{\prime }}\right)}{\Delta t} \end{array}\right)\\ = & -i\int dx\bar{\chi }\left(x\right)\;\lim_{\Delta t\to 0}\left(\begin{array}{c} \frac{\chi_{{}_{0+\Delta }}}{\Delta t}\\ \frac{\chi_{t+\Delta }-\chi_{t}}{\Delta t} \end{array}\right)\\ = & -i\int dx\bar{\chi }\left(x\right)\;\left(\begin{array}{c} \left\{\begin{array}{cc} 0 & \;\mathrm{if}\;\chi_{0}=0\\ \infty & \mathrm{otherwise} \end{array}\right.\\ \partial_{t}\chi^{\prime } \end{array}\right) \end{array}$$(86)$$\begin{array}{cc} = & -i\int dx\bar{\chi }\left(x\right)\;\partial_{t}\chi \left(x\right) \end{array}$$(87)$$\begin{array}{cc} = & -i{\bar{\chi }}^{†}\partial_{t}\chi \end{array}$$(88)$$\begin{array}{cc} B= & i\;\int \;dx^{\prime }\beta \left(x^{\prime }\right)\;\int \;dx\;\frac{\delta \bar{\chi }\left(x\right)}{\delta \bar{\chi }\left(x^{\prime }\right)}\partial_{t}\varphi \left(x\right)\\ = & i\;\int \;dx^{\prime }\beta \left(x^{\prime }\right)\;\int \;dx\;\delta \left(x-x^{\prime }\right)\partial_{t}\varphi^{\prime }\left(x\right)\\ = & i\;\int \;dx\;\int \;dx^{\prime }\;\beta \left(x^{\prime }\right)\delta \left(x-x^{\prime }\right)\partial_{t}\varphi \left(x\right)\\ = & i\;\int \;dx\;\beta \left(x\right)\partial_{t}\varphi \left(x\right)\\ = & i\beta^{†}\partial_{t}\varphi \end{array}$$ such that,
(89)$$\begin{array}{cc} H(\psi |\varphi_{0})\; & \hat{=}\;i{\bar{\chi }}^{†}\partial_{t}\chi -i\beta^{†}\partial_{t}\varphi +i\left\{Q,\bar{Q}\right\}. \end{array}$$

The crucial insight is given by Equation  (Equation 85). If $\chi_{0}\neq 0,$ the expression *A* would diverge and Equation  (Equation 83) would not hold. In order to reestablish a compact notation in Equation (Equation 86), we note that any finite assignment of $\partial_{t}\chi_{0}\neq 0$ would only make a vanishing contribution to the integral as being on an infinitesimal smart support.

The information Hamiltonian of Equation  (Equation 83) has two parts. We call the left part, which contains the time derivatives of the fermionic and bosonic fields, the dynamic information. The right part, which is described by the Poisson bracket, is referred to as the static information. The derivation of Poisson brackets in a system with fermionic and bosonic fields is described in [38,39].

This yields the partition function,
(90)$$\begin{array}{c} Z\propto \int D\psi \;e^{-i{\bar{\chi }}^{†}\partial_{t}\chi +i\beta^{†}\partial_{t}\varphi -i\left\{Q,\bar{Q}\right\}-H\left(\varphi_{0}\right)}. \end{array}$$

So far we represented the partition function in terms of the signal field, φ, and the three fields, $\beta ,\;\chi ,\;\bar{\chi }$.

In case of a white excitation field ξ, the partition function of DFI can be derived using the Markov property. For this, we start with the IFT partition function for a bosonic field φ and a fermionic field χ and decompose it in terms of time-ordered conditional probabilities:(91)$$\begin{array}{cc} \; & Z=\int D\varphi \;D\chi \;P(\varphi ,\chi )\\ \; & =\left(\prod_{n=0}^{N}\int D\varphi_{n}\;\int D\chi_{n}\right)\;P(\varphi_{N},\chi_{N},\varphi_{N-1},\chi_{N-1}, \end{array}$$(92)$$\begin{array}{cc} \; & \;\;\;\varphi_{N-2},...,\varphi_{1},\chi_{1},\varphi_{0}),\\ & =\left(\prod_{n=0}^{N}\int D\varphi_{n}\;\int D\chi_{n}\right)\;P\left(\varphi_{N},\chi_{N}|\varphi_{N-1},\chi_{N-1}\right) \end{array}$$(93)$$\begin{array}{cc} \; & \;\;\times .......\times P\left(\varphi_{1},\chi_{1}|\varphi_{0}\right)P\left(\varphi_{0}\right) \end{array}$$
where $\varphi_{0}=\varphi \left(\cdot ,t_{0}\right)$ is the field at initial time $t_{0}=0$ while there is no $\chi_{0}=\chi \left(\cdot ,t_{0}\right)$ .

The conditional probabilities can then be represented as QFT transition amplitudes [40,41] between states of the system denoted by the Dirac notation as:(94)$$\begin{array}{ccc} P(\varphi_{k},\;\chi_{k}|\varphi_{j},\;\chi_{j}) & =: & \langle \varphi_{k},\chi_{k},t_{k}\left|\right|\varphi_{j},\chi_{j},t_{j}\rangle \\ & := & \langle \varphi_{k},\chi_{k}|M(t_{k},t_{j})|\varphi_{j},\chi_{j}\rangle . \end{array}$$

At this stage, these are formal definitions, with the time localized states $\langle \varphi_{k},\chi_{k},t_{k}|:=\delta \left(\varphi \left(\cdot ,t_{k}\right)-\varphi_{k}\right)\;\delta \left(\chi \left(\cdot ,t_{k}\right)-\chi_{k}\right)$, $|\varphi_{j},\chi_{j},t_{j}\rangle :=\delta \left(\varphi \left(\cdot ,t_{j}\right)-\varphi_{j}\right)\;\delta \left(\chi \left(\cdot ,t_{j}\right)-\chi_{j}\right)$, and the not localized ones $\langle \varphi_{k},\chi_{k}|:=\delta \left(\varphi \left(\cdot ,t\right)-\varphi_{k}\right)\;\delta \left(\chi \left(\cdot ,t\right)-\chi_{k}\right)$, $|\varphi_{j},\chi_{j}\rangle :=\delta \left(\varphi \left(\cdot ,t\right)-\varphi_{j}\right)\;\delta \left(\chi \left(\cdot ,t\right)-\chi_{j}\right)$, with *t* being some unspecified time. Here, *j* and *k* label time-slice field configurations, like $\varphi \left(\cdot ,t\right)=\varphi_{j}$ and $\varphi \left(\cdot ,t\right)=\varphi_{k}$, and their associated times are $t=t_{j}$ and $t=t_{k}.$ The first line does not contain a usual scalar product between states, as the variables have first to be brought to a common time. This is done in the second line by the transfer operator $M(t_{k},t_{j})$, which describes the mapping of states at time $t_{j}$ to such at $t_{k}$. In [19], it is shown that a representation of these state vectors is given by the exterior algebra over the field configuration space.

By assigning field operators to the fermionic and bosonic fields, χ and φ, as well as their momenta, ν and ω, respectively, the partition function in Equation  (Equation 93) can be rewritten in terms of the generalized Fokker–Planck operator of the states $\hat{H}$ according to [31,40,41,42]. $\hat{H}$ is not to be confused with the information Hamiltonian $H(\psi |\varphi_{0})$. The precise relation of these will be established in the following.

As mentioned in [18,19,20,21], the time evolution operator $\hat{H}$ is not Hermitian and thus the time evolution is not described by the Schrödinger equation but by the generalized Fokker–Planck equation instead:(95)$$\begin{array}{ccc} \partial_{t}|\varphi ,\chi ,t\rangle & = & -\hat{H}|\varphi ,\chi ,t\rangle \end{array}$$(96)$$\begin{array}{ccc} \Rightarrow |\varphi ,\chi ,t+\Delta t\rangle & = & e^{-\hat{H}\Delta t}|\varphi ,\chi ,t\rangle \end{array}$$(97)$$\begin{array}{ccc} \Rightarrow M(t_{k},t_{j}) & = & e^{-\hat{H}\left(t_{k}-t_{j}\right)}. \end{array}$$

These and the following equations define the properties of Ĥ. The conditional probabilities for the fields $\varphi_{k}$ and $\chi_{k}$, given the fields at the previous time step $\varphi_{k-1}$, $\chi_{k-1}$ are given by the transition amplitudes between the corresponding states and are defined via the time evolution:(98)$$\begin{array}{ccc} P_{k,k-1} & = & P(\varphi_{k},\chi_{k}|\varphi_{k-1},\chi_{k-1})\\ & = & \langle \varphi_{k},\chi_{k},t_{k}\left|\right|\varphi_{k-1},\chi_{k-1},t_{k-1}\rangle \\ & = & \langle \varphi_{k},\chi_{k}\left|e^{-\hat{H}\Delta t}\right|\varphi_{k-1},\chi_{k-1}\rangle . \end{array}$$

At this point we multiply with unity,
(99)$$\begin{array}{ccc} 𝟙 & = & \int D\omega_{k}\;D\nu_{k}\;|\omega_{k},\nu_{k}\rangle \langle \omega_{k},\nu_{k}|, \end{array}$$
where the $|\omega_{k},\nu_{k}\rangle$ are momentum eigenstates of the field that obey on equal time slices:(100)$$\begin{array}{ccc} \langle \omega_{k},\nu_{k}|\varphi_{k},\chi_{k}\rangle & = & e^{-i\omega_{k}\varphi_{k}+i\nu_{k}\chi_{k}}. \end{array}$$

If we choose infinitesimal small time steps, we can evaluate the time-evolution operator on the momentum eigenstate, which leads to the following expression for the conditional probability:(101)$$\begin{array}{ccc} P_{k,k-1} & = & \int D\omega_{k}\;D\nu_{k}\;\langle \varphi_{k},\chi_{k}|e^{-\hat{H}\Delta t}\\ & & \times |\omega_{k},\nu_{k}\rangle \langle \omega_{k},\nu_{k}|\varphi_{k-1},\chi_{k-1}\rangle \\ & = & \int D\omega_{k}\;D\nu_{k}\;e^{-i\omega_{k}\varphi_{k-1}+i\nu_{k}\chi_{k-1}}\\ & & \times \langle \varphi_{k},\chi_{k}\left|e^{-\hat{H}\Delta t}\right|\omega_{k},\nu_{k}\rangle \\ & \propto & \int D\omega_{k}\;D\nu_{k}\;e^{-H\left(\varphi_{k},\chi_{k},\omega_{k},\nu_{k}\right)\Delta t-i\omega_{k}\varphi_{k-1}}\\ & & \times e^{+i\nu_{k}\chi_{k-1}}\langle \varphi_{k},\chi_{k}|\omega_{k},\nu_{k}\rangle \\ & = & \int D\omega_{k}D\nu_{k}e^{-H\left(\varphi_{k},\chi_{k},\omega_{k},\nu_{k}\right)\Delta t+i\omega_{k}\left(\varphi_{k}-\varphi_{k-1}\right)}\\ & & \times e^{-i\nu_{k}\left(\chi_{k}-\chi_{k-1}\right)}. \end{array}$$

The formal definition of $H(\varphi_{k},\chi_{k},\omega_{k},\nu_{k})$ for Δt→0 is:(102)$$H\left(\varphi_{k},\chi_{k},\omega_{k},\nu_{k}\right)=-\frac{1}{\Delta t}\ln\langle \varphi_{k},\chi_{k}\left|e^{-\hat{H}\Delta t}\right|\omega_{k},\nu_{k}\rangle .$$

With this in mind the conditional transition probability distributions can be written in terms of the function *H*. In the next step, these are inserted into the partition function in Equation (Equation 92). Taking the limit Δt→0, N→∞ leads to:(103)$$\begin{array}{cc} Z & \propto \int D\psi \;e^{-\int dt\;H\left(\varphi_{t},\chi_{t},\omega_{t},\nu_{t}\right)+i\omega^{†}\partial_{t}\varphi -i\nu^{†}\partial_{t}\chi -H\left(\varphi_{0}\right)}. \end{array}$$

In the end, the partition function in Equation (Equation 90) needs to be equal to the partition function in Equation (Equation 103) in order to guarantee consistency of the theory. This permits the following identifications,
(104)$$\begin{array}{cc} \nu & =\bar{\chi }, \end{array}$$
(105)$$\begin{array}{cc} \omega & =\beta , \end{array}$$
(106)$$\begin{array}{cc} \int dt\;H(\psi_{t}) & =i\{Q\left(\psi \right),\bar{Q}\left(\psi \right)\}. \end{array}$$

To sum up, it was shown that the auxiliary fields $\bar{\chi }$ and β are simply the momenta of the ghost field χ and the signal field φ, respectively. And, for the moment the more important finding is that the time evolution is governed by the *Q*-exact static information, i.e., $\int dt\;H\left(t\right)=i\left\{Q,\bar{Q}\right\}$. Comparing Equation  (Equation 89) to Equation  (Equation 106), we find this enters directly the information Hamiltonian,
(107)$$H\left(\psi |\varphi_{0}\right)\;\hat{=}\;i{\bar{\chi }}^{†}\partial_{t}\chi -i\beta^{†}\partial_{t}\varphi +\int dt\;H\left(\psi_{t}\right),$$
which can be regarded in combination with Equation (Equation 80) as the central connection between STS and IFT, relating the information Hamiltonian $H(\psi |\varphi_{0})$ for the full system trajectory to the Fokker–Planck evolution operators $H(\psi_{t})$ on individual time-slices. H is a dimensionless quantity, whereas *H* has the units of a rate.

In [19] it is shown that {Q,·} is the path-integral version of the exterior derivative $\hat{d}$ in the exterior algebra. This recognition allows to identify the time-evolution in Equation (Equation 106) as the path-integral version of the time-evolution operator in the Focker–Planck equation. Moreover, it is demonstrated that this time-evolution operator is $\hat{d}$-exact and since the exterior derivative is nilpotent, the exterior derivative commutes with the time-evolution. The conclusion is made that this corresponds to a topological supersymmetry. Firstly, $\hat{d}$ as the operator representative of {Q,·} interchanges fermions and bosons, since it replaces one bosonic field variable by a fermionic one. Secondly, since a physical system is symmetric with regard to an operator, if the operator commutes with the time-evolution operator. As this is the case for $\hat{d}$ and $\hat{H}$, the field dynamics is supersymmetric.

Here it should be recalled that the ghost fields are scalar with fermionic statistics. In thise sense, the symmetry generated by the charge *Q* can be considered as a Becchi-Rouet-Stora-Tyutin (BRST) symmetry [43] in the context of this paper. Still, for further investigations of STS in [18,19], the formulation of the generated symmetry as a topological supersymmetry according to [44] is crucial. For this reason, we talk about a topological supersymmetry in this paper).

### 4.4. Spontaneous SUSY Breaking and Field Inference

The supersymmetry of a dynamical field can be spontaneously broken [18,19,20,21]. This coincides with the appearance of dynamical chaos as characterized by positive Lyaponov exponents for the growth of the difference of nearby system trajectories. It is intuitively clear that the occurrence of chaos will reduce the predictability of the system and therefore make field inference from measurements more difficult. We hope that the here established connection of DFI and STS will permit to quantify the impact of chaos on field inference in future research. For the time being, we investigate the reverse impact, that of measurements on the supersymmetry of the field knowlege as encoded in the partition function.

## 5. SUSY and Measurements

### 5.1. Abstract Considerations

In Section 2.6, we introduced the moment generating function in IFT in order to calculate field expectation values after measurement data *d* became available. For a dynamical field, this can now be written with the help of STS according to Equation (Equation 29) as:(108)$$\begin{array}{ccc} Z_{d}\left[J\right] & \;=\; & \int D\varphi \;e^{-H\left(d,\varphi \right)+J^{†}\varphi }\\ & \;=\; & \int D\varphi P\left(\varphi \right)P\left(d|\varphi \right)e^{J^{†}\varphi }\\ & \;=\; & \int D\varphi ∥\frac{\delta G[\varphi ]}{\delta \varphi^{\prime }}∥G\left(G\left[\varphi \right],\Xi \right)P\left(\varphi_{0}\right)P\left(d|\varphi \right)e^{J^{†\varphi }}\\ & \;\propto \; & \int D\psi e^{\left\{Q,-{\bar{\chi }}^{†}G\left[\varphi \right]-\frac{1}{2}{\bar{\chi }}^{†}\Xi \beta \right\}-H\left(\varphi_{0}\right)-H\left(d|\varphi \right)+J^{†}\varphi }. \end{array}$$

Note that we removed the −i factor from the Fermionic variables that was introduced in Equation (Equation 61) in order to connect to the conventions of the STS literature. Doing so alleviates the necessity to take the absolute value from the corresponding term. From Equation (Equation 108), we see that the combined information representing the knowledge from measurement data *d* and about the dynamics as expressed by the θ-function from Equation (Equation 74) consists of several parts,
(109)$$\begin{array}{ccc} H(d,\psi ) & \hat{=} & \left\{Q,\theta \left(\psi \right)\right\}+H\left(d|\varphi \right)+H\left(\varphi_{0}\right)\\ & = & -{\bar{\chi }}^{†}\partial_{t}\chi -i\beta^{†}\partial_{t}\varphi^{\prime }+\left\{Q\left(\psi \right),\bar{Q}\left(\psi \right)\right\}\\ & & +H\left(d|\varphi \right)+H\left(\varphi_{0}\right). \end{array}$$

The first part, $-{\bar{\chi }}^{†}\partial_{t}\chi -i\beta^{†}\partial_{t}\varphi^{\prime }+\left\{Q\left(\psi \right),\bar{Q}\left(\psi \right)\right\},$ describes the dynamics of the field $\varphi^{\prime }$ and that of the ghost fields χ and $\bar{\chi }$ for times after the initial moment by a *Q*-exact term, meaning that supersymmetry is conserved if only this would affect the fields for non-inital times $t>t_{0}$. The last term, $H\left(\varphi_{0}\right)=-\ln P\left(\varphi_{0}\right)$, describes our knowledge on the initial conditions and not of the evolving field. The middle term, H(d|φ)=−lnP(d|φ), describes the knowledge gain by the measurement. If it addresses non-inital times, it is in general not *Q*-exact. Thus, if one would take the perspective of including the measurement constraints into the system dynamics, as it was done with the noise excitation, the thereby extended system would not be *Q*-exact any more. The reason for this is that “external forces” need to be introduced into the system description to guide its evolution through the constraints set by the measurement, which are not stationary and Gaussian as the excitation noise is. Or more precisely, the knowledge state on the excitation field ξ is in general not a zero-centered Gaussian prior with a stationary correlation structure any more, but a posterior P(ξ|d) with explicitly time-dependent mean and correlation structure in ξ.

### 5.2. Idealized Linear Dynamics

In order to illustrate the impact of chaos on the predictability of a system, we analyze a simplified, but instructive scenario. Our starting point is the information Hamiltonian for all fields, Equation (Equation 109), integrated over the β field,
(110)$$\begin{array}{ccc} & & H(d,\varphi ,\chi ,\bar{\chi })\\ & = & -\ln\int D\beta \;e^{-H(d,\psi )}\\ & = & -\ln\int D\beta \;e^{-\left\{Q,\theta \left(\psi \right)\right\}-H\left(d|\varphi \right)-H\left(\varphi_{0}\right)}\\ & = & -\ln\int D\beta \;e^{i\beta^{†}G\left[\varphi \right]-\frac{1}{2}\beta^{†}\Xi \beta -{\bar{\chi }}^{†}G^{\prime }\left[\varphi \right]\;\chi -H\left(d|\varphi \right)-H\left(\varphi_{0}\right)}\\ & \hat{=} & -\ln e^{-\frac{1}{2}G{\left[\varphi \right]}^{†}\Xi^{-1}G\left[\varphi \right]-{\bar{\chi }}^{†}G^{\prime }\left[\varphi \right]\;\chi -H\left(d|\varphi \right)-H\left(\varphi_{0}\right)}\\ & = & \frac{1}{2}G{\left[\varphi \right]}^{†}\Xi^{-1}G\left[\varphi \right]+{\bar{\chi }}^{†}G^{\prime }\left[\varphi \right]\;\chi +H\left(d|\varphi \right)+H\left(\varphi_{0}\right). \end{array}$$

The information Hamiltonian contains now, in this order, terms that represent the excitation noise statistics G(ξ,Ξ) (as ξ=G[φ]), the functional determinant of the dynamics (represented with help of fermionic fields), the measurement information H(d|φ), and the information on the initial condition $H(\varphi_{0})$.

We assume the system φ to be initially $\varphi \left(\cdot ,0\right)=\varphi_{0}$ at t=0 and to obey Equation (Equation 34) afterwards with ξ↩G(ξ,𝟙), i.e., Ξ=𝟙. We can then define a classical field $\varphi_{\mathrm{cl}}$ that obeys the excitation-free dynamics:(111)$$\partial_{t}\varphi_{\mathrm{cl}}\left(x\right)=F\left[\varphi_{\mathrm{cl}}\right]\left(x\right)$$
and a deviation $\varepsilon :=\varphi -\varphi_{\mathrm{cl}}$ from this, which evolves according to:(112)$$\begin{array}{ccc} \varepsilon (\cdot ,0) & = & 0\;\mathrm{and}\\ \partial_{t}\varepsilon & = & F\left[\varphi_{\mathrm{cl}}+\varepsilon \right]-F\left[\varphi_{\mathrm{cl}}\right]+\xi \end{array}$$(113)$$\begin{array}{ccc} & = & \underset{=:A}{\underbrace{\frac{\partial F[\varphi_{\mathrm{cl}}]}{\partial \varphi_{\mathrm{cl}}}}}\;\varepsilon +\xi +O\left(\varepsilon^{2}\right). \end{array}$$

Here, we performed a first-order expansion in the deviation field. Furthermore, we assume that only a sufficiently short period after t=0 is considered, such that second-order effects in ε as well as any time dependence of *A* can be ignored. For this period, we have the solution:(114)$$\varepsilon_{t}=\int_{0}^{t}d\tau \;e^{A(t-\tau )}\xi_{t^{\prime }}.$$

Further, we imagine that a system measurement at time $t=t_{o}$ probes perfectly a normalized eigendirection *b* of *A*, i.e., that we get noiseless data according to:(115)$$d=R\varphi =b^{\hat{†}}\varepsilon \left(\cdot ,t_{o}\right).$$

Here, $R_{1\;(\vec{x},t)}:=b_{\vec{x}}\delta \left(t-t_{o}\right)$ is the linear measurement operator, *b* fulfills:(116)$$\begin{array}{ccc} A\;b & = & \lambda_{b}\;b, \end{array}$$
with $\lambda_{b}$ the corresponding eigenvalue, and $\hat{†}$ denoting the adjoint with respect to spatial coordinates only. $\lambda_{b}$ is also the Lijapunov coefficient of the dynamical mode *b*, which is stable for $\lambda_{b}<0$ and unstable for $\lambda_{b}>0$. The latter is a prerequisite for chaos.

Finally, to exclude any further complications, we assume that *A* can be fully expressed in terms of a set of such orthonormal eigenmodes,
(117)$$A=\sum_{a}\lambda_{a}\;a\;a^{\hat{†}}\;\mathrm{with}\;a^{\hat{†}}a^{\prime }=\delta_{aa^{\prime }}.$$

Now, we are in a convenient position to work out our knowledge on ε for all times for which our idealizing assumptions hold.

A priori, the deviation evolves with an average:(118)$${\bar{\varepsilon }}_{t}:={\langle \varepsilon_{t}\rangle }_{\left(\xi \right)}=\int_{0}^{t}d\tau \;e^{A(t-\tau )}\underset{=0}{\underbrace{{\langle \xi_{\tau }\rangle }_{\left(\xi \right)}}}=0$$
and an dispersion, most conveniently expressed in the eigenbasis of *A*, of: 
(119)$$\begin{array}{ccc} E_{\left(a,t\right)\left(a^{\prime },t^{\prime }\right)} & := & {\langle a^{\hat{†}}\varepsilon_{t}\varepsilon_{t^{{}^{\prime }}}^{\hat{†}}a^{\prime }\rangle }_{\left(\xi \right)}\\ & = & \int_{0}^{t}d\tau \;\int_{0}^{t^{\prime }}d\tau^{\prime }\;a^{\hat{†}}e^{A(t-\tau )}\times \\ & & \underset{=\delta (\tau -\tau^{\prime })\;𝟙}{\underbrace{{\langle \xi_{\tau }\xi_{\tau^{\prime }}^{\hat{†}}\rangle }_{\left(\xi \right)}}}e^{A^{\hat{†}}\left(t^{\prime }-\tau^{\prime }\right)}a^{\prime }\\ & = & \int_{0}^{\min(t,t^{\prime })}d\tau \;\;e^{\lambda_{a}\left(t-\tau \right)}a^{\hat{†}}a^{\prime }\;e^{\lambda_{a^{\prime }}\left(t^{\prime }-\tau \right)}\\ & = & e^{\lambda_{a}\left(t+t^{\prime }\right)}\delta_{aa^{\prime }}\left[1-e^{\lambda_{a}\min\left(t,t^{\prime }\right)}\right]\;{\left(2\lambda_{a}\right)}^{-1}\\ & = & \delta_{aa^{\prime }}\underset{=:f_{a}\left(t,t^{\prime }\right)}{\underbrace{\left[e^{\lambda_{a}\left(t+t^{\prime }\right)}-e^{\lambda_{a}\left|t-t^{\prime }\right|}\right]{\left(2\lambda_{a}\right)}^{-1}}}\;. \end{array}$$

We introduced here with $f_{a}\left(t,t^{\prime }\right):={\langle a^{\hat{†}}\varepsilon_{t}\varepsilon_{t^{{}^{\prime }}}^{\hat{†}}a\rangle }_{\left(\xi \right)}$ the a priori temporal correlation function of a field eigenmode *a*. Since both the dynamics as well as the measurement keep the eigenmodes separate in our illustrative example, we only obtain additional information on the mode *b* from our measurement. This is given according to Equation (Equation 33) by the posterior:(120)$$\begin{array}{ccc} P(\varepsilon |d) & = & G(\varepsilon -m,D) \end{array}$$
with posterior mean:(121)$$m=E\;R^{†}{\left(RE\;R^{†}\right)}^{-1}d$$
and posterior uncertainty:(122)$$D=E-E\;R^{†}{\left(RE\;R^{†}\right)}^{-1}R\;E,$$
which follow respectively from Equations (Equation 27) and (Equation 23) for the limit of vanishing noise covariance *N*. Expressing these in the eigenbasis of *A* gives:(123)$$\begin{array}{ccc} m_{a}\left(t\right) & := & a^{\hat{†}}m\left(\cdot ,t\right)\\ & = & \delta_{ab}\frac{f_{b}\left(t,t_{o}\right)}{f_{b}\left(t_{o},t_{o}\right)}\;d \end{array}$$
and
(124)$$\begin{array}{ccc} D_{\left(a,t\right)\left(a^{\prime },t^{\prime }\right)} & = & \delta_{aa^{\prime }}\left[f_{a}\left(t,t^{\prime }\right)-\delta_{ab}\frac{f_{b}\left(t,t_{o}\right)f_{b}\left(t^{\prime },t_{o}\right)}{f_{b}\left(t_{o},t_{o}\right)}\right]. \end{array}$$

Figure 1 shows the mean and uncertainty dispersion of the measured mode for various values of $\lambda_{b}$. The correlation between different modes $a\neq a^{\prime }$ vanishes and therefore any mode a≠b behaves like a prior mode shown in grey in Figure 1. For the measured mode *b*, the propagator is in general non-zero, but vanishes for times separated by the observation, e.g., $D_{\left(b,t\right)\left(b,t^{\prime }\right)}=0$ for $t<t_{o}<t^{\prime }$, as one can easily verify:(125)$$\begin{array}{ccc} & & D_{\left(b,t\right)\left(b,t^{\prime }\right)}\times f_{b}\left(t_{o},t_{o}\right)\\ & = & f_{b}\left(t,t^{\prime }\right)\;f_{b}\left(t_{o},t_{o}\right)-f_{b}\left(t,t_{o}\right)f_{b}\left(t^{\prime },t_{o}\right)\\ & = & \left[e^{\lambda_{b}\left(t+t^{\prime }\right)}-e^{\lambda_{b}\left|t-t^{\prime }\right|}\right]\;\left[e^{2\lambda_{b}t_{o}}-1\right]-\\ & & \left[e^{\lambda_{b}\left(t+t_{o}\right)}-e^{\lambda_{b}\left|t-t_{o}\right|}\right]\;\left[e^{\lambda_{b}\left(t_{o}+t^{\prime }\right)}-e^{\lambda_{b}\left|t_{o}-t^{\prime }\right|}\right]\\ & = & \left[e^{\lambda_{b}\left(t+t^{\prime }\right)}-e^{\lambda_{b}\left(t^{\prime }-t\right)}\right]\;\left[e^{2\lambda_{b}t_{o}}-1\right]-\\ & & \left[e^{\lambda_{b}\left(t+t_{o}\right)}-e^{\lambda_{b}\left(t_{o}-t\right)}\right]\;\left[e^{\lambda_{b}\left(t_{o}+t^{\prime }\right)}-e^{\lambda_{b}\left(t^{\prime }-t_{o}\right)}\right]\\ & = & \left[e^{\lambda_{b}\left(t+t^{\prime }+2t_{o}\right)}-e^{\lambda_{b}\left(t+t^{\prime }\right)}-e^{\lambda_{b}\left(t^{\prime }-t+2t_{o}\right)}+e^{\lambda_{b}\left(t^{\prime }-t\right)}\right]-\\ & & \left[e^{\lambda_{b}\left(t+t^{\prime }+2t_{o}\right)}-e^{\lambda_{b}\left(t+t^{\prime }\right)}-e^{\lambda_{b}\left(2t_{o}+t^{\prime }-t\right)}+e^{\lambda_{b}\left(t^{\prime }-t\right)}\right]\\ & = & 0. \end{array}$$

Thus, the perfect measurement introduces a so-called Markov-blanket, which separates the periods before and after it from each other. Knowing anything above earlier times than $t_{o}$ does not inform about later times, as the measurement at $t_{o}$ provides the only relevant constraint for the later period. The equal time uncertainty of the measured mode is:(126)$$\begin{array}{ccc} D_{\left(b,t)(b,t\right)} & = & f_{b}\left(t,t\right)-\frac{f_{b}\left(t,t_{o}\right)f_{b}\left(t,t_{o}\right)}{f_{b}\left(t_{o},t_{o}\right)}\\ & = & \frac{1}{2\lambda_{b}}\left[e^{2\lambda_{b}t}-1-\frac{{\left[e^{\lambda_{b}\left(t+t_{o}\right)}-e^{\lambda_{b}\left|t-t_{o}\right|}\right]}^{2}}{2\lambda_{b}\left[e^{2\lambda_{b}t_{o}}-1\right]}\right]. \end{array}$$

Figure 1 shows this for a number of instructive values of $\lambda_{b}.$ The impact of the Liapunov exponent on the predictability of the system is clearly visible. The larger the Liapunov exponent, the faster the uncertainties grow. This can be seen by comparison of the top panels or by inspection of the bottom middle panel of Figure 1. Thus, chaos, which implies the existence of positive Liapunov exponents, makes field inference more difficult. This, however, is only true on an absolute scale. If one considers relative uncertainties, as also displayed in Figure 1 on the bottom right, then it turns out that these grow slowest for the more unstable modes. This is the memory effect of chaotic systems, which can remember small initial disturbances for long, if not infinite times.

To simplify the system further, we concentrate first on the case $\lambda_{b}=0$, which corresponds to a Wiener process. For this we get:(127)$$\begin{array}{ccc} f_{a}\left(t,t^{\prime }\right) & = & \min(t,t^{\prime }), \end{array}$$
implying a posterior mean of: (128)$$\begin{array}{ccc} m_{a}\left(t\right) & = & \delta_{ab}\min\left(t\;/\;t_{o},1\right)\;d \end{array}$$
and an information propagator of:(129)$$\begin{array}{ccc} D_{\left(a,t\right)\left(a^{\prime },t^{\prime }\right)} & = & \delta_{aa^{\prime }}\left[\min\left(t,t^{\prime }\right)-\delta_{ab}\frac{\min\left(t,t_{o}\right)\min\left(t^{\prime },t_{o}\right)}{t_{o}}\right]. \end{array}$$

This provides the equal time uncertainty for our measured mode *b*: (130)$$\begin{array}{ccc} D_{\left(b,t)(b,t\right)} & = & t-\frac{\min{\left(t,t_{o}\right)}^{2}}{t_{o}}=\left\{\begin{array}{cc} t\;(1-t\;/\;t_{o}) & t<t_{o}\\ t-t_{o} & t\geq t_{o} \end{array}\right.,\;\;\;\;\;\;\;\;\;\; \end{array}$$
which is also shown in Figure 1 in both middle panels. This scenario with $\lambda_{b}=0$ corresponds to a Wiener process, which sits on the boundary between the stable Ornstein–Uhlenbeck process with $\lambda_{b}<0$ and the instability of chaos with $\lambda_{b}>0$. This marginal stable case should now be taken into the non-linear regime.

### 5.3. Idealized Non-Linear Dynamics

We saw that the posterior uncertainty is a good indicator for the difficulty to predict the field at locations or times where or when it was not measured. This holds—modulo some corrections—also in the case of non-linear dynamics, which introduces non-Gaussianities into the field statistics.

In order to investigate such a non-Gaussian example, we extend the previous case with $\lambda_{b}=0$ to the next order in ε, while still assuming that all modes are dynamically decoupled (up to that order), such that we only need to concentrate on the dynamics of $\varepsilon_{b}\left(t\right):=b^{\hat{†}}\varepsilon \left(\cdot ,t\right),$
(131)$$\partial_{t}\varepsilon_{b}=\frac{1}{2}\;\mu_{b}\;\varepsilon_{b}^{2}+\xi +O\left(\varepsilon_{b}^{3}\right),$$
where again $\hat{†}$ denotes an integration in position space only. This mode will exhibit an infinite posterior mean for times larger than $t_{o}$. To understand why, let us first investigate the noise free solution of $\partial_{t}\varepsilon_{b}=\frac{1}{2}\;\mu_{b}\;\varepsilon_{b}^{2}$ for some finite starting value $\varepsilon \left(t_{i}\right)=\varepsilon_{i}$ at $t_{i}>t_{o}$. This might have been created by an excitation fluctuation during the period $\left[t_{o},t_{i}\right]$ for which always a potentially tiny, but finite probability exists. The free solution after $t_{i\;}$ is given by:(132)$$\varepsilon_{b}\left(t\right)=\frac{\varepsilon_{i}}{1-\frac{1}{2}\;\varepsilon_{i}\;\mu_{b}\left(t-t_{i}\right)},$$
which develops a singularity for $\varepsilon_{i}\;\mu_{b}>0$ in the finite period $\tau =2/(\varepsilon_{i}\;\mu_{b})$. Thus, there is a finite probability that at time $t_{s}=t_{i}+\tau$ the system is at infinity, and this lets also the expectation value of ε diverge for $t_{s}$. This moment, when the expectation value has diverged, can be made arbitrarily close to $t_{o}$, as the Gaussian fluctuations in ξ permit to reach any necessary $\varepsilon_{i}$ at say $t_{i}=\left(t_{s}-t_{o}\right)/2=\tau$ with a small, but finite probability, where $\varepsilon_{i}=2/\left(\tau \;\mu_{b}\right)=4/\left[\left(t_{s}-t_{o}\right)\;\mu_{b}\right]$.

For times $t\in [0,t_{o}]$, in between the moments when the two data points were measured, the posterior mean should stay finite. The reason is that any a priori possible trajectory diverging to (plus) infinity (for $\mu_{b}>0$) during this period is excluded a posteriori by the data point $\left(t,\varepsilon_{b}\right)=\left(1,1\right)$. Such trajectories could not have taken place, as the dynamics does not permit trajectories to return from (positive) infinite values to finite ones, since that would require an infinite large (negative) excitation, which does have a probability of zero.

Let us assume that for the period $t\in [0,t_{o}]$, the second-order approximation of the dynamical equation holds. We then have:(133)$$\begin{array}{ccc} G[\varepsilon_{b}] & = & \partial_{t}\varepsilon_{b}-\frac{1}{2}\;\mu_{b}\;\varepsilon_{b}^{2}, \end{array}$$
and therefore,
(134)$$\begin{array}{ccc} \frac{\delta G[\varepsilon_{b}]}{\delta \varepsilon_{b}} & = & \partial_{t}-\mu_{b}\;\varepsilon_{b}. \end{array}$$

Inserting this into Equation (Equation 111) yields:(135)$$\begin{array}{ccc} H(d,\varphi ,\chi ,\bar{\chi }) & = & \frac{1}{2}{\left[\partial_{t}\varepsilon_{b}-\frac{1}{2}\;\mu_{b}\;\varepsilon_{b}^{2}\right]}^{†}\left[\partial_{t}\varepsilon_{b}-\frac{1}{2}\;\mu_{b}\;\varepsilon_{b}^{2}\right]\\ & & +{\bar{\chi }}^{†}\left[\partial_{t}-\mu_{b}\;\varepsilon_{b}\right]\;\chi +H\left(d|\varphi \right)+H\left(\varphi_{0}\right).\\ & = & H_{\mathrm{free}}\left(d,\varphi ,\chi ,\bar{\chi }\right)+H_{\mathrm{int}}\left(d,\varphi ,\chi ,\bar{\chi }\right) \end{array}$$
with
(136)$$\begin{array}{ccc} H_{\mathrm{free}}\left(d,\varphi ,\chi ,\bar{\chi }\right) & = & \frac{1}{2}\varepsilon_{b}^{†}\partial_{t}^{†}\partial_{t}\varepsilon_{b}+{\bar{\chi }}^{†}\partial_{t}\;\chi +H\left(d|\varphi \right)+H\left(\varphi_{0}\right)\\ H_{\mathrm{int}}\left(d,\varphi ,\chi ,\bar{\chi }\right) & = & -\mu_{b}{\left[\varepsilon_{b}^{2}\right]}^{†}\partial_{t}\;\varepsilon_{b}+\frac{\mu_{b}^{2}}{8}{\left[\varepsilon_{b}^{2}\right]}^{†}\left[\varepsilon_{b}^{2}\right]\\ & & -\mu_{b}{\bar{\chi }}^{†}\left(\varepsilon_{b}\;\chi \right). \end{array}$$

The free information Hamiltonian $H_{\mathrm{free}}\left(d,\varphi ,\chi ,\bar{\chi }\right)$ defines the Wiener process field inference problem we addressed before, and has the classical field as well as the bosonic and fermionic propagators given by:

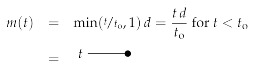
(137)

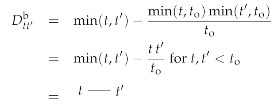
(138)

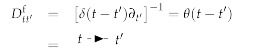
(139)
respectively. Here, we introduced their Feynman diagram representation as well. The Fermionic propagator is the inverse of $\delta \left(t-t^{\prime }\right)\partial_{t^{\prime }}$ as is verified by:(140)$$\begin{array}{ccc} \int dt^{\prime }\;\left[\delta \left(t-t^{\prime }\right)\partial_{t^{\prime }}\right]D_{t^{\prime }t^{\prime \prime }}^{f} & = & \int dt^{\prime }\;\left[\delta \left(t-t^{\prime }\right)\partial_{t^{\prime }}\right]\theta \left(t^{\prime }-t^{\prime \prime }\right)\\ & = & \int dt^{\prime }\;\delta \left(t-t^{\prime }\right)\delta \left(t^{\prime }-t^{\prime \prime }\right)\\ & = & \delta \left(t-t^{\prime \prime }\right)={𝟙}_{tt^{\prime \prime }}. \end{array}$$

The interacting Hamiltonian $H_{\mathrm{int}}\left(d,\varphi ,\chi ,\bar{\chi }\right)$ provides the following interaction vertices: 
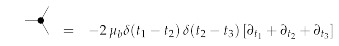
(141)

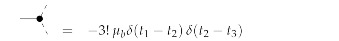
(142)

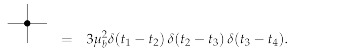
(143).

The integration over the time axis in Feynman diagrams can be restricted to the interval $\left[0,t_{o}\right]$ as the propagator vanishes for (exactly) one of the times being larger than $t_{o}$, see Equation (Equation 125).

To first order in $\mu_{b}$, the posterior mean and uncertainty dispersion for $0\leq t,t^{\prime }\leq t_{o}$ are then given by the Feynman diagrams:

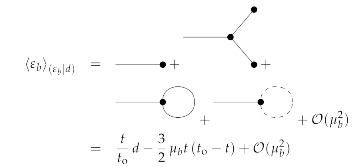
(144)

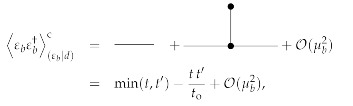
(145)
see Appendix A. It turns out that all first-order diagrams (in $\mu_{b})$ with a bosonic three-vertex are zero. The reason for this lies in the fact that these are all of a similar form,
(146)$$\begin{array}{ccc} & & \int_{0}^{t_{o}}dt_{1}\int_{0}^{t_{o}}dt_{2}\int_{0}^{t_{o}}dt_{3}\delta \left(t_{1}-t_{2}\right)\;\delta \left(t_{2}-t_{3}\right)\times \\ & & \left[D_{tt_{3}}^{b}\partial_{t_{1}}g\left(t_{1},t_{2}\right)+D_{tt_{3}}^{b}\partial_{t_{2}}g\left(t_{1},t_{2}\right)+g\left(t_{1},t_{2}\right)\partial_{t_{3}}D_{tt_{3}}^{b}\right]\\ & = & \int_{0}^{t_{o}}dt_{1}\left[D_{tt_{1}}^{b}\partial_{t_{1}}g\left(t_{1},t_{1}\right)+g\left(t_{1},t_{1}\right)\partial_{t_{1}}D_{tt_{1}}^{b}\right]\\ & = & \int_{0}^{t_{o}}dt_{1}D_{tt_{1}}^{b}\partial_{t_{1}}g\left(t_{1},t_{1}\right)\\ & & +{\left[g\left(t_{1},t_{1}\right)D_{tt_{1}}^{b}\right]}_{t_{1}=0}^{t_{o}}-\int_{0}^{t_{o}}dt_{1}D_{tt_{1}}^{b}\partial_{t_{1}}g\left(t_{1},t_{1}\right)\\ & = & g\left(t_{o},t_{o}\right)D_{t_{o}t_{o}}^{b}-g\left(0,0\right)D_{00}^{b}=0, \end{array}$$
with $g\left(t_{1},t_{2}\right)=\mu_{b}m_{t_{1}}m_{t_{2}},\;\frac{1}{2}\mu_{b}D_{t_{1}t_{2}}^{b}$, and $\mu_{b}m_{t_{1}}D_{t_{2}t^{\prime }}^{b}$ respectively. All these diagrams vanish, because $D_{t_{o}t_{o}}^{b}=D_{00}^{b}=0$. Thus, to first order in $\mu_{b}$ only a correction due to the Fermionic loop is necessary. This is negative (for positive $\mu_{b}$) as from the sum over trajectories, which go through the initial data $\left(t_{i},\varepsilon_{bi}\right)=\left(0,0\right)$ as well as through the later observed data $\left(t_{o},\varepsilon_{bo}\right)=\left(1,1\right)$, all the trajectories that diverge prematurely (within $t\in [0,t_{o}]$) are excluded.

The posterior mean and uncertainty of the scenario with $\lambda_{b}=0$ and $\mu_{b}=0.3$ is displayed for $t\in [0,t_{o}]$ in the middle panel of Figure 2 in red in comparison to those for $\lambda_{b}=0$ and $\mu_{b}=0$ in cyan. It can there be observed that the exclusion of the diverging trajectories by the observation has made the ensemble of remaining trajectories stay away from high values, which more easily diverge. Furthermore, this effect is solely represented by the fermionic Feynman diagram, as all bosonic corrections vanish (for $\lambda_{b}=0$) up to the considered linear order in $\mu_{b}$. Thus, taking the functional determinant into account, for which the fermionic fields were introduced, is important in order to arrive at the correct posterior statistics. This effect naturally arises in the used Stratonovich formalism of stochastic systems, and is less obvious in Îto’s formalism.

Now, we are in a position to also work out the corrections in case λ≠0. In this case, we have:(147)$$\begin{array}{ccc} G[\varepsilon_{b}] & = & \partial_{t}\varepsilon_{b}-\lambda_{b}\;\varepsilon_{b}-\frac{1}{2}\;\mu_{b}\;\varepsilon_{b}^{2} \end{array}$$
and
(148)$$\begin{array}{ccc} \frac{\delta G[\varepsilon_{b}]}{\delta \varepsilon_{b}} & = & \partial_{t}-\lambda_{b}-\mu_{b}\;\varepsilon_{b} \end{array}$$
such that now:(149)$$\begin{array}{ccc} H_{\mathrm{free}}\left(d,\varphi ,\chi ,\bar{\chi }\right) & = & \frac{1}{2}\varepsilon_{b}^{†}\left[{\left(\partial_{t}-\lambda_{b}\right)}^{†}\left(\partial_{t}-\lambda_{b}\right)\right]\varepsilon_{b}\\ & & +{\bar{\chi }}^{†}\left(\partial_{t}-\lambda_{b}\right)\;\chi +H\left(d|\varphi \right)+H\left(\varphi_{0}\right)\\ H_{\mathrm{int}}\left(d,\varphi ,\chi ,\bar{\chi }\right) & = & -\mu_{b}{\left[\varepsilon_{b}^{2}\right]}^{†}\left(\partial_{t}-\lambda_{b}\right)\;\varepsilon_{b}+\frac{\mu_{b}^{2}}{8}{\left[\varepsilon_{b}^{2}\right]}^{†}\left[\varepsilon_{b}^{2}\right] \end{array}$$(150)$$\begin{array}{ccc} & & -\mu_{b}{\bar{\chi }}^{†}\left(\varepsilon_{b}\;\chi \right) \end{array}$$
and

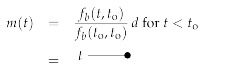
(151)

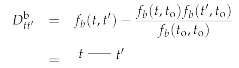
(152)

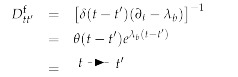
(153)
where:(154)$$\begin{array}{ccc} f_{b}\left(t,t^{\prime }\right) & = & \frac{e^{\lambda_{b}\left(t+t^{\prime }\right)}-e^{\lambda_{b}\left|t-t^{\prime }\right|}}{2\lambda_{b}}. \end{array}$$

The Fermionic propagator for λ≠0 is easily verified:(155)$$\begin{array}{ccc} & & \int dt^{\prime }\delta \left(t-t^{\prime }\right)\left(\partial_{t^{\prime }}-\lambda_{b}\right)D_{tt^{\prime }}^{f}\\ & = & \int dt^{\prime }\delta \left(t-t^{\prime }\right)\left(\partial_{t^{\prime }}-\lambda_{b}\right)\left[\theta \left(t^{\prime }-t^{\prime \prime }\right)e^{\lambda_{b}\left(t^{\prime }-t^{\prime \prime }\right)}\right]\\ & = & \int dt^{\prime }\delta \left(t-t^{\prime }\right)\left[\delta \left(t^{\prime }-t^{\prime \prime }\right)e^{\lambda_{b}\left(t^{\prime }-t^{\prime \prime }\right)}\right.\\ & & \left.+\theta \left(t^{\prime }-t^{\prime \prime }\right)\lambda_{b}e^{\lambda_{b}\left(t^{\prime }-t^{\prime \prime }\right)}-\lambda_{b}\theta \left(t^{\prime }-t^{\prime \prime }\right)e^{\lambda_{b}\left(t^{\prime }-t^{\prime \prime }\right)}\right]\\ & = & \delta (t-t^{\prime \prime }). \end{array}$$

The only changed interaction vertex is:

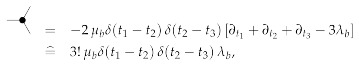
(156)
where we used in the last step that the derivatives lead to vanishing contribution to all diagrams up to linear order in $\mu_{b}$ as we showed in Equation (Equation 146). The relevant diagrams correcting the posterior mean are then:

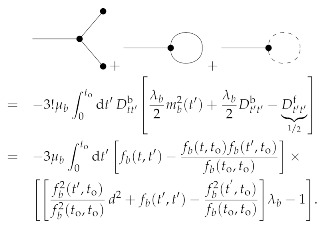
(157)

This integral can be calculated analytically. However, the resulting expression is relatively complicated, therefore omitted here, and only plotted in Figure 2. We calculate it with the computer algebra system SymPy [45]. The same is true for the first order (in $\mu_{b}$) correction to the uncertainty:

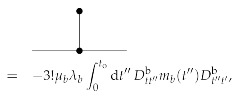
(158)
which we also only present graphically in Figure 2.

This figure shows that in all displayed cases ($\lambda_{b}\in \left\{-1,0,1\right\}$), the posterior trajectories preferentially avoid getting close to easily diverging regimes (larger positive values for $\mu_{b}>0$), and they avoid such areas more, as more linear dynamics are unstable (i.e., larger values of $\lambda_{b}$).

Interestingly, the interplay of this non-linear dynamics with the constraint provided by the measurement leads to a reduced a posteriori uncertainty for unstable systems ($\lambda_{b}>0$) for times prior to the measurement. This is not in contradiction to the notion of chaotic systems being harder to predict. Here, we are looking at trajectories that could have lead—starting from some known value—to the observed situation at a later time. Thanks to the stronger divergence of trajectories of chaotic systems, the variety of trajectories that pass through both the initial condition and the later observed situation, is smaller than if the system is not chaotic. Thus, the measurement provides more information for this period in the chaotic regime, but less for the period after the measurement.

## 6. Conclusions and Outlook

We brought dynamical field inference based on information field theory and the suspersymmetric theory of stochastics into contact. To this end, we showed that the DFI partition function becomes the STS one if the excitation of the field becomes white Gaussian noise and no measurements constrain the field evolution. In this case, the dynamical system has a supersymmetry. We note that neither STS nor DFI are limited to the white noise case.

For chaotic systems, this supersymmetry is broken spontaneously. As the presence of chaos limits the ability to predict a system, DFI for systems with broken supersymmetry should become more difficult. We hope that the here established connection of STS and DFI allows to quantitatively investigate this.

While re-deriving basic elements of STS within the framework of IFT, we carefully investigated the domains on which the different fields and operators live and act, respectively, using the perspective that the continuous time description of the system should be the limiting case of a discrete time representation for vanishing time steps. Thereby, we showed, for example, that the fermionic ghost field has to vanish on the initial time slice for the theory to be consistent.

Furthermore, we showed that most measurements of the field during its evolution phase do not obey the system’s supersymmetry, and are not *Q*-exact. Nevertheless, the formalism of STS is still applicable and might help to develop advanced DFI schemes. For example, two of the challenges DFI is facing are the representation of the dynamics enforcing delta function and a Jacobian in the path integral of the DFI partition function. For these, STS introduces bosonic Lagrange and fermionic ghost fields. Using those in perturbative calculations, for example via Feynman diagrams, might allow to develop DFI schemes that are able to cope with non-linear dynamical systems.

In order to illustrate how such a non-linear dynamics inference would look like, we investigate a simplified situation, in which the deviation of a system driven by stochastic external excitation from the classical (not perturbed system) is measured at an initial and a later time. The simplifications we impose are that (i) the measurement probes exactly one eigenmode of the linear part of the evolution operator for these deviations, that (ii) the evolution operator stays stationary during the considered period (thus different modes do not mix), and that the non-linear part of the evolution is also (iii) stationary, (iv) second order in the observed eigenmode, and (v) keeps that mode also separate from the other modes (no non-linear mode mixing). Under these particular conditions (i)–(v), the field inference problem becomes a one dimensional problem for the measured mode as a function of time, which can be treated exactly for a vanishing non-linearity and perturbatively with the help of Feynman diagrams in case of non-vanishing non-linearity. Thereby, it turns out that the Fermionic contributions, which implement the effect of the functional determinant, are key to obtain the correct a posteriori mean of the system.

The investigation of the illustrative example show a few things. First, predicting the future evolution of a more chaotic system from measurements is harder than for a less chaotic one as the absolute uncertainty of the measured mode increases faster in the former situation. This is not very surprising, but the following insight might be: The relative uncertainty (uncertainty standard deviation over absolute value of the deviation) grows slower for a chaotic system. This is an echo of the known memory effect of chaotic systems, which remember small perturbations in unstable modes for a longer time thanks to their rapid amplification. Third, non-linear dynamics, which can lead to even more drastic divergence of system trajectories (even to infinity in finite times), makes prediction of the future even harder, but enhances the amount of information measurements provide for periods between them. Due to the larger sensitivity of the system to perturbations, the measurements now exclude more trajectories that were possible a priori.

Thus, the interplay of measurements and non-linear chaotic systems is complex and more interesting phenomena should become visible as soon as the simplifying assumptions (i)–(v) made in our illustrative example are dropped. For those, the inclusion of the Fermionic part of the information field theory of stochastic systems will be as essential to obtain the correct statistics on the system trajectories as it is in our idealized illustrative example. We believe that insights provided by the stochastic theory of supersymmetry will continue to pay off in investigations of more complex systems, which we leave for future research.

## Figures and Tables

**Figure 1 entropy-23-01652-f001:**
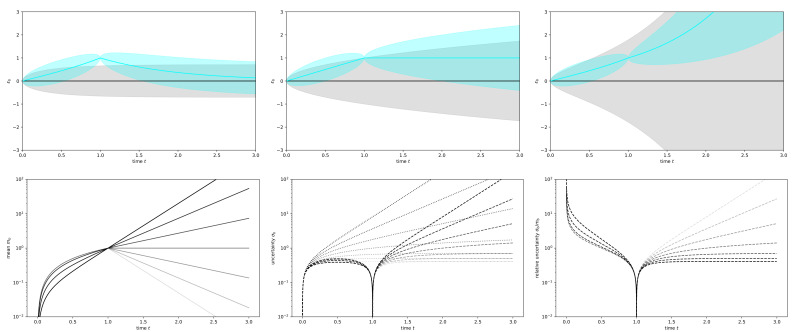
Illustration of the knowledge on a measured system mode b. Top row: A priori (gray) and a posteriori (cyan) field mean (lines) and one sigma uncertainty (shaded) for an Ornstein–Uhlenbeck process (left, $\lambda_{b}=-1)$, a Wiener process (middle, $\lambda_{b}=0)$, and a chaotic process (right, $\lambda_{b}=1$) of a system eigenmode *b* after one perfect measurement at $t_{o}=1$. Bottom row: The same, but on logarithm scales and for Liapunov exponents $\lambda_{b}=-3$, −2, −1, 0, 1, 2, and 3, as displayed in colors ranging from light to dark gray in this order (i.e., strongest chaos is shown in black). Left: Posterior mean. Middle: Uncertainty of prior (dotted) and posterior (dashed). Right: Relative posterior uncertainty.

**Figure 2 entropy-23-01652-f002:**
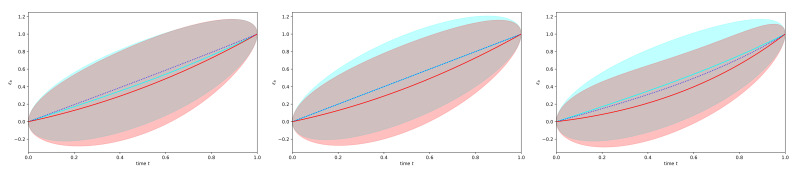
Like top row of Figure 1 just for the non-linear system defined by Equation (Equation 147) within the period t∈[0,1] with first-order bosonic and fermionic perturbation corrections for $\mu_{b}=0.3$ in red, as in Figure 1 without such non-linear corrections in cyan, and with only bosonic corrections in blue (dotted, displayed without uncertainty). The three panels display the cases $\lambda_{b}=-1$ (left), $\lambda_{b}=0$ (middle), and $\lambda_{b}=1$ (right). Note that the a priori mean and uncertainty dispersion are both infinite for any time t>0, as without the measurement, trajectories reaching positive infinity within finite times are not excluded from the ensemble of permitted possibilities.

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
