# Peer review of "Dynamical Field Inference and Supersymmetry"

_entropy, 2021, doi:10.3390/e23121652_

Round 1

Reviewer 1 Report

This is an interesting paper. It explores the possibility to represent DFI (dynamical field inference) by means of a field theory, called IFT (information field theory). One of the remarkable thing is the use of ghost fields in order to represent field determinants that show up inside the partition function when dealing with non-free stochastic systems. Another remarkable aspect is the appearance of supersymmetry (STS). At this point supersymmetry breaking is identified with the onset of chaos. The scheme is very interesting. I recommend the publication of this paper. I have only a few minor remarks:
1) before eq.(35) I have not been able to spot the definition of the space C^{n-1} where the operator G acts on.
2) I don't think the use of the verb `marginalize' before eq.(110) is standard; there are other examples of a peculiar use of English, which should perhaps be amended.

Reviewer 2 Report

This article is a review in Information Theory of Fields and has many intersections with a previous publication of one of the authors. However,  adds some new aspects: the inclusion of ghost fields to rewrite a functional determinant, the introduction of supersymmetry, and propose the study of chaotic systems using spontaneously broken supersymmetry.   The first part is quite elementary from the point of view of Quantum Field Theory (QFT) since most of the arguments are included in basic references of QFT. As for me, I would not call the symmetry generated by the charge Q a supersymmetry; instead, I would call it a BRST symmetry (Becchi-Rouet-Stora-Tyutin) and to Q a BRST charge (see http://www.scholarpedia.org/article/Becchi-Rouet-Stora-Tyutin_symmetry). In the sense that the ghost fields are scalar fields with fermionic statistics, then in that sense, the symmetry does not map strictly between bosonic-fermionic fields. Furthermore, the delta functional included in  (5) can be considered as a first-class constraint in Dirac's sense (see Henneaux, M.; Teitelboim, C. Quantization of gauge Systems; Princeton University Press: Princeton, NJ, USA, 1992. ).  In general, the article is quite illustrative, and if the authors make the appropriate changes, I will consider accepting the present manuscript.
